# Ciliary Rab28 and the BBSome negatively regulate extracellular vesicle shedding

**Jyothi S Akella[1‡]\*, Stephen P Carter[2‡]\*, Ken Nguyen[3], Sofia Tsiropoulou[2], Ailis L Moran[2], Malan Silva[1,4], Fatima Rizvi[1], Breandan N Kennedy[2], David H Hall[3], Maureen M Barr[1†]\*, Oliver E Blacque[2†]\***

[1]Department of Genetics and Human Genetics Institute of New Jersey, Rutgers University, Piscataway, United States; [2]School of Biomolecular and Biomedical Science, Conway Institute, University College Dublin, Dublin, Ireland; [3]Center for *C. elegans* Anatomy, Albert Einstein College of Medicine, Bronx, United States; [4]Department of Biology, University of Utah, Salt Lake City, United States

**\*For correspondence:**
akella@dls.rutgers.edu (JSA);
stephen.carter@ucdconnect.ie (SPC);
barr@dls.rutgers.edu (MMB);
oliver.blacque@ucd.ie (OEB)

[†]These authors contributed equally to this work
[‡]These authors also contributed equally to this work

**Competing interests:** The authors declare that no competing interests exist.

**Abstract** Cilia both receive and send information, the latter in the form of extracellular vesicles (EVs). EVs are nano-communication devices that influence cell, tissue, and organism behavior. Mechanisms driving ciliary EV biogenesis are almost entirely unknown. Here, we show that the ciliary G-protein Rab28, associated with human autosomal recessive cone-rod dystrophy, negatively regulates EV levels in the sensory organs of *Caenorhabditis elegans* in a cilia specific manner. Sequential targeting of lipidated Rab28 to periciliary and ciliary membranes is highly dependent on the BBSome and the prenyl-binding protein phosphodiesterase 6 subunit delta (PDE6D), respectively, and BBSome loss causes excessive and ectopic EV production. We also find that EV defective mutants display abnormalities in sensory compartment morphogenesis. Together, these findings reveal that Rab28 and the BBSome are key in vivo regulators of EV production at the periciliary membrane and suggest that EVs may mediate signaling between cilia and glia to shape sensory organ compartments. Our data also suggest that defects in the biogenesis of cilia-related EVs may contribute to human ciliopathies.

## Introduction

Cilia are conserved microtubule (MT)-based organelles that extend from the surfaces of most eukaryotic cell types. Cilia serve a variety of functions that include motility and signal transduction, as well as the capacity to shed extracellular vesicles (EV) (*Wang and Barr, 2018*; *Carter and Blacque, 2019*; *Nachury and Mick, 2019*). Defects in primary cilia lead to monosymptomatic and syndromic human disorders, collectively termed ciliopathies. These include diseases such as autosomal dominant polycystic kidney disease (ADPKD), retinitis pigmentosa, cone-rod dystrophy and Bardet-Biedl syndrome (BBS) (*Waters and Beales, 2011*).

Cilia depend on several modes of intracellular transport to establish and control their molecular make-up (*Jensen and Leroux, 2017*). Intraflagellar transport (IFT) is driven by kinesin-II anterograde (ciliary base to tip) and cytoplasmic dynein 2 retrograde (ciliary tip to base) motors (*Rosenbaum and Witman, 2002*). Cargo adapters such as the IFT-A/B complexes and the BBSome enable IFT to transport structural and signaling proteins into and out of cilia (*Liem et al., 2012*; *Bhogaraju et al., 2013*; *Lechtreck, 2015*; *Nachury, 2018*). The BBSome regulates ciliary signaling cascades by removing various membrane proteins from cilia via retrograde IFT (*Lechtreck et al., 2009*; *Liu and Lechtreck, 2018*; *Ye et al., 2018*) and consists of an octameric complex (BBS1/2/4/5/7/8/9 and BBS18) that is recruited to membranes by the small G-protein Arl6 (*Liew et al., 2014*). Lipidated membrane proteins are shuttled into cilia via prenyl- (e.g PDE6D) and myristoyl- (eg. UNC119) binding proteins,

and subsequently released from these shuttles via cargo-displacement factors such as Arl3 (*Ismail et al., 2012*; *Fansa et al., 2016*; *Jensen and Leroux, 2017*).

In addition to its role as a signaling hub, the cilium is also an evolutionarily conserved site of extracellular vesicle (EV) biogenesis and shedding (*Wood et al., 2013*; *Wang et al., 2014a*; *Salinas et al., 2017*). Most EV studies have used cultured cells or biofluids, and much effort has been invested in improving methodology for isolating and characterizing different EV subtypes using protein and lipid markers. Exosomes (30–150 nm in diameter) are generated through fusion of multi-vesicular bodies. Microvesicles or ectosomes (100–1000 nm) bud from the plasma membrane (*Maas et al., 2017*; *Raposo and Stoorvogel, 2013*; *Meldolesi, 2018*). EVs are heterogeneous and typically isolated based on size and density and are of unknown biogenic origin: available markers are not strictly specific for exosomes or microvesicles/ectosomes. Therefore, the terms small EVs (<200 nm) and large EVs (>200 nm) are typically used, and the term 'small EV' encompasses both exosomes and small ectosomes/microvesicles (*Théry et al., 2018*). To overcome these challenges in the EV field, we developed a genetically-encoded fluorescent-protein tagged EV cargo tracking system that enables study of ciliary EV biogenesis, shedding, and release in living animals (*Wang et al., 2014a*; *Wang et al., 2015*; *Maguire et al., 2015*; *Silva et al., 2017*; *O'Hagan et al., 2017*).

EVs released from the tips of cilia are clearly ectosomal in origin (*Wood et al., 2013*; *Phua et al., 2017*). Several functions have been proposed for ciliary EVs, including signaling between cells and/or organisms (*Wang et al., 2014a*; *Cao et al., 2015*), waste disposal (*Nager et al., 2017*), ciliary resorption and disassembly (*Long et al., 2016*; *Phua et al., 2017*), and morphogenesis of elaborate ciliary membranes (*Salinas et al., 2017*). Defects in EV biogenesis and release are associated with ciliary dysfunction (*Wang et al., 2015*; *Maguire et al., 2015*; *Wang et al., 2014a*; *O'Hagan et al., 2017*; *Silva et al., 2017*; *Dilan et al., 2018*). Abnormal EV phenotypes have also been observed in ciliopathies such as polycystic kidney disease (PKD), polycystic liver disease, and retinal degeneration (*Hogan et al., 2009*; *Salinas et al., 2017*; *Dilan et al., 2018*; *Masyuk et al., 2010*). However, the relationship between ciliary EVs and cilia function remains unknown.

The *C. elegans* nematode is an excellent in vivo system to study ciliary formation, specialization, and function including EV shedding (*Bargmann, 2006*). In *C. elegans,* sensory cilia are present at the dendritic tips of sensory neurons, with most housed within sensory organs (sensilla). Most of these organs are environmentally exposed via cuticular pores derived from surrounding glial cell processes (*Singhvi and Shaham, 2019*). Of the 302 neurons that comprise the hermaphrodite nervous system, 60 are ciliated (*Ward et al., 1975*; *Perkins et al., 1986*; *Doroquez et al., 2014*). *C. elegans* males possess an additional 54 ciliated neurons (*Sulston et al., 1980*; *Cook et al., 2019*; *Molina-García et al., 2019*). *C. elegans* cilia possess a similar basic structural plan, extending from a degenerate basal body into a transition zone (TZ), and an axonemal structure that displays tremendous diversity in terms of their MT arrangements between different cell-types (*Perkins et al., 1986*; *Doroquez et al., 2014*; *Silva et al., 2017*; *Akella et al., 2019*). At the distal-most end of the dendrite, the periciliary membrane compartment (PCMC) acts as a trafficking hub for sorting proteins to and from the cilium (*Kaplan et al., 2012*; *Nechipurenko et al., 2017*; *Serwas et al., 2017*).

The cephalic male (CEM) and shared inner labial (IL2) neurons are the only known ciliated EV-releasing neurons (EVNs) as defined by imaging of fluorescently-tagged EV cargos released outside the animal (*Wang et al., 2014a*). These environmentally-released EVs promote inter-animal communication (*Wang et al., 2014a*; *Silva et al., 2017*). In *C. elegans*, EVs are 'shed' at the ciliary base and retained within lumens of the cephalic and inner labial sensory organs as visualized by transmission electron microscopy (TEM) and tomography reconstructions. EVs are also 'released' from cilia directly into the environment outside of the sensory organs, as observed by fluorescently-tagged EV cargos such as the TRP polycystin PKD-2::GFP and myristoylated peripheral membrane protein CIL-7 (*Wang et al., 2014a*; *Maguire et al., 2015*). Ciliary EV release is controlled by the tubulin code writers, erasers and readers (*Wang et al., 2014a*; *Wang et al., 2015*; *Maguire et al., 2015*; *Silva et al., 2017*; *O'Hagan et al., 2017*). In these mutants, EVs are shed at the ciliary base and accumulate in the lumens of sensory organs but are not released into the environment. Unlike the ciliary EV release machinery, the EV ciliary base shedding machinery remains poorly characterized.

Small G-proteins including those of the Rab family regulate cilium formation and function, as well as EV biogenesis (*Li and Hu, 2011*; *Blacque et al., 2018*; *Knödler et al., 2010*; *Blanc and Vidal, 2018*; *Meldolesi, 2018*). Here we focus on Rab28, which is associated with autosomal recessive cone-rod dystrophy (arCRD) (*Roosing et al., 2013*; *Riveiro-Álvarez et al., 2015*; *Lee et al., 2017*).

Photoreceptor degeneration is also found in many ciliopathies, including Bardet-Biedl syndrome (*Waters and Beales, 2011*). In *C. elegans* amphid and phasmid cilia, RAB-28 is a BBS-8-dependent IFT cargo that associates with the periciliary membrane (PCM) when in a GTP-bound conformation (*Jensen et al., 2016*). Functionally, *C. elegans* RAB-28 disruption leads to defects in amphid sensory compartment size via a proposed cell non-autonomous mechanism, previously speculated to involve EVs (*Jensen et al., 2016*). An EV-related function for RAB-28 is also suggested by its significant overrepresentation in a *C. elegans* EVN transcriptome (*Wang et al., 2015*). Whether or not RAB-28 plays a role in EV biology, and how RAB-28 functions in cilia is unknown.

Here we dissect the transport pathway that regulates *C. elegans* RAB-28 localization in non-EVN (amphid/phasmid neurons) and EVN cilia. We find that RAB-28's PCM association is dependent on the BBSome and, to a lesser extent, its regulator ARL-6, whilst its IFT transport requires the BBSome, PDL-1 (PDE6D ortholog) and ARL-3. We also identify cell-specific distinctions in how RAB-28 transport machinery is deployed, with PDL-1 dispensable for maintaining RAB-28's PCM levels in amphid/phasmid neurons but not EVNs. Functional analyses revealed that BBSome mutants phenocopy the amphid sensory compartment morphogenesis defects observed in RAB-28-disrupted worms. Importantly, loss of RAB-28 or BBSome genes, but not PDL-1, causes ectopic EV accumulation in the cephalic EVN sensory organ, suggesting that RAB-28 and the BBSome negatively regulate EV biogenesis and/or shedding at the PCM. This research reveals a BBSome-ARL-6-PDL-1 network for targeting RAB-28, a critical EV regulator, to specific ciliary membranes, thus allowing regulation of EVs at distinct sites. The members of this transport network as well as RAB-28 and the EV cargo PKD-2 are all well-established ciliary proteins associated with several human genetic diseases (*Waters and Beales, 2011*; *Reiter and Leroux, 2017*). We propose that dysregulated ciliary EV production may be a factor in the pathology of certain ciliopathies (*Wang and Barr, 2016*).

## Results

### RAB-28 transport to and within amphid and phasmid cilia is regulated by the BBSome, ARL-6 and prenyl-binding PDL-1

In amphid and phasmid neurons, we previously showed that the IFT motility and periciliary membrane (PCM) targeting of GFP-tagged RAB-28$^{Q95L}$, a putative GTP-preferring and active form of the G-protein, is dependent on the BBSome component *bbs-8* (*Jensen et al., 2016*). To further establish the role of the BBSome in RAB-28 ciliary localization and transport we assessed GFP-RAB-28$^{Q95L}$ in *bbs-5(gk507)* null mutant cilia. We also assessed GFP-RAB-28$^{Q95L}$ localization in a null mutant of *arl-6*, which regulates membrane-targeting of the BBSome in mammalian cells (*Jin et al., 2010*). We found that GFP-RAB-28$^{Q95L}$ mislocalization in the amphid and phasmid cilia of *bbs-5(gk507)* worms is indistinguishable from *bbs-8(nx77)* mutants; GFP signals are diffused throughout the neurons, with no detectable PCM enrichment or IFT movement (*Figure 1*, enhanced contrast images and expressivity shown in *Figure 1—figure supplement 1A and C*). By contrast, *arl-6(ok3472)* mutants display a weaker phenotype, with modestly reduced levels of RAB-28 at the PCM and a diffuse dendritic localization (*Figure 1*). Also, in *arl-6* phasmid cilia, the frequency of GFP-RAB-28$^{Q95L}$- positive IFT trains is increased twofold (*Figure 1—figure supplement 1B* and *Video 1*). These data indicate that a complete and properly regulated BBSome is required for normal targeting and retention of RAB-28 to the PCM and that ARL-6 inhibits RAB-28 association with IFT trains, at least in amphid and phasmid neurons.

As RAB-28 is a prenylated protein, we investigated whether RAB-28 ciliary targeting requires the *C. elegans* orthologs of PDE6d (PDL-1) and ARL3 (ARL-3). We found that whilst the PCM localization of GFP-RAB-28$^{Q95L}$ remains intact, the reporter's IFT motility is completely lost in *pdl-1(gk157)* and *arl-3(1703)* null mutants (*Figure 1* and *Figure 1—figure supplement 1C*). An additional punctate localization was also detected for GFP-RAB-28$^{Q95L}$ in the phasmid cell bodies of *pdl-1(gk157)* mutants. Thus, like the BBSome, the lipidated protein shuttle PDL-1 is also required for RAB-28 association with IFT trains in amphid and phasmid neurons, although it is dispensable for RAB-28 targeting to the PCM of these cells.

To explore the genetic relationship between the BBSome, *arl-6* and *pdl-1*, we assessed RAB-28 localization in *pdl-1(gk157);bbs-8(nx77)* and *pdl-1(gk157);arl-6(ok3472)* double mutants. GFP::RAB-28$^{Q95L}$ localization and IFT behavior in *pdl-1(gk157);bbs-8(nx77)* mutants was found to be identical

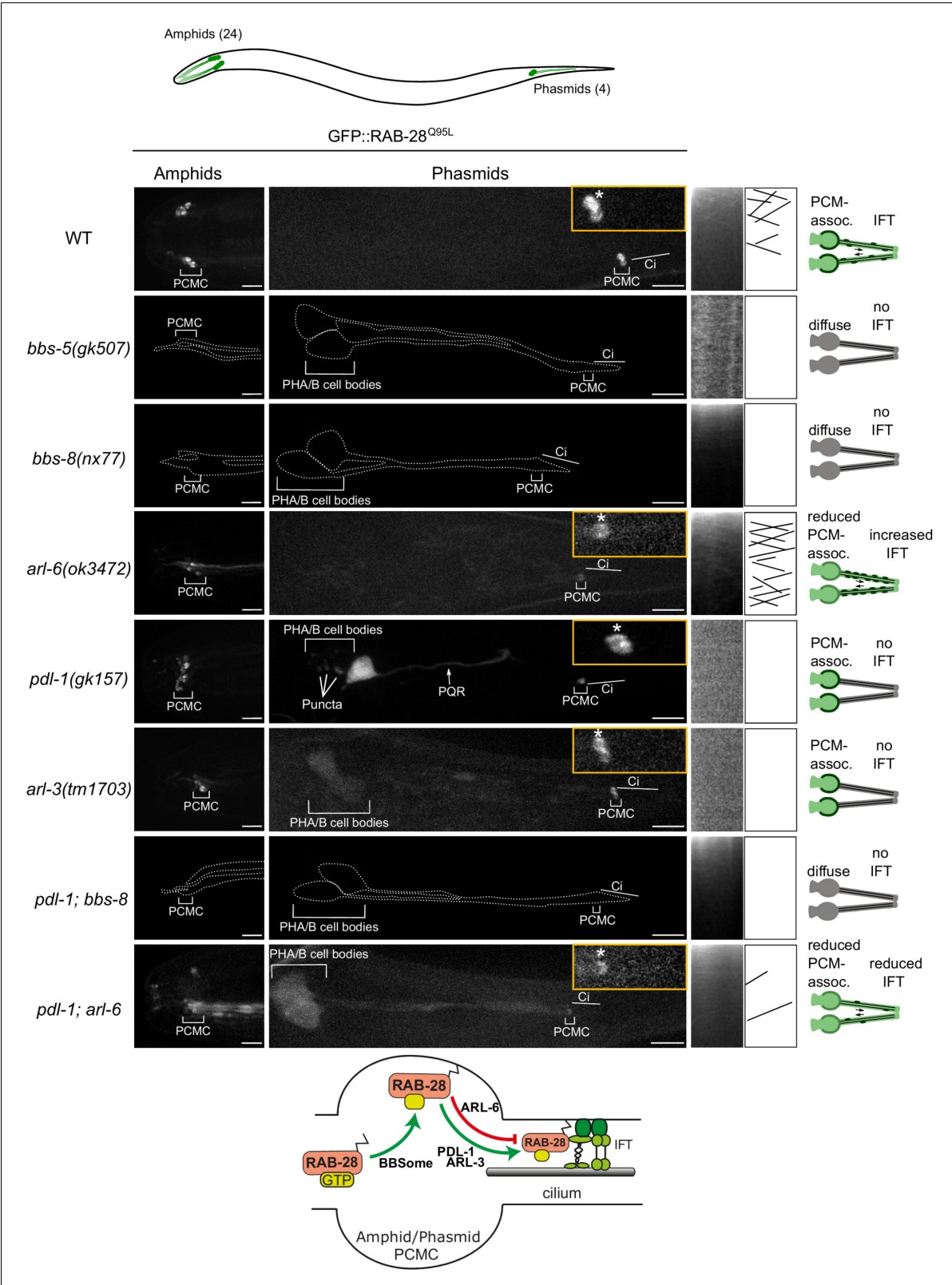

**Figure 1.** A BBSome-ARL-6-PDL-1 network targets RAB-28 to sensory cilia. Representative confocal z-projection images of amphid (head) and phasmid (tail) neurons from hermaphrodites of the indicated genotype expressing GFP::RAB-28$^{Q95L}$. Anterior is to the left; all images taken at identical exposure settings. Traced outlines in *bbs-5*, *bbs-8* and *pdl-1;bbs-8* panels are derived from intensity-adjusted images (see *Figure 1—figure supplement 1A*). Insets; higher magnification images of phasmid cilia, with PCMC denoted by asterisks. Kymograph x-axis represents distance and y-axis time (scale

*Figure 1 continued on next page*

*Figure 1 continued*

bars; 5 s and 1 μm), and both retrograde and anterograde particle lines are shown in the kymograph schematics. Schematics on the right summarize the phenotypes observed in a pair of phasmid cilia. Bottom schematic shows proposed model for RAB-28 transport to amphid and phasmid channel neuronal cilia. ci: cilium; PQR: additional ciliated neuron in the tail that occasionally expresses the RAB-28 reporter.

The online version of this article includes the following source data and figure supplement(s) for figure 1:

**Figure supplement 1.** GFP::RAB-28 localization and IFT frequency analysis.
**Figure supplement 1—source data 1.** Data for *Figure 1* and *Figure 1—figure supplement 1B*.

to that of *bbs-8(nx77)* single mutants, reproducing the complete loss of RAB-28's PCM association and IFT in that strain (*Figure 1* and enhanced contrast images in *Figure 1—figure supplement 1A*), indicating that *bbs-8* is epistatic (masking) to *pdl-1*. Similarly, in *pdl-1(gk157);arl-6(ok3472)* worms, the reduced GFP::RAB-28$^{Q95L}$ PCM localization phenocopies that of the *arl-6(ok3472)* single mutant, although there are additional diffuse signals in the cilium and distal dendrite (*Figure 1*). With regard to RAB-28's IFT behavior phenotype, *pdl-1* loss strongly suppresses the increased IFT frequency phenotype associated with *arl-6* disruption, further supporting the critical role of PDL-1 in regulating RAB-28 loading onto IFT trains (*Figure 1—figure supplement 1B,C* and *Video 1*). It is notable, however, that a low level of RAB-28 IFT movement remains in at least some *pdl-1(gk157);arl-6(ok3472)* double mutant cilia (*Figure 1—figure supplement 1B,C* and *Video 1*). Since detectable IFT was never observed in the *pdl-1(gk157)* single mutant, the latter suggests that *arl-6* and *pdl-1* mutations are mutually suppressive, consistent with an opposing relationship in regulating the formation of RAB-28-positive IFT trains.

Taken together, these data reveal a ciliary trafficking pathway in amphid and phasmid neurons whereby activated RAB-28 is initially targeted to the PCM by the BBSome, and to a lesser extent ARL-6, and subsequently solubilized by PDL-1 for loading onto IFT trains, a step inhibited by ARL-6 (summarized in *Figure 1* schematic).

## A modified BBSome-ARL-6-PDL-1 network targets RAB-28 to the periciliary membrane and cilia of extracellular vesicle-releasing neurons (EVNs)

Having identified a BBSome-PDL-1 ciliary targeting pathway for RAB-28 in amphid and phasmid channel neurons, we explored RAB-28 ciliary trafficking in extracellular vesicle-releasing neurons (EVNs) because: (i) we speculated previously that the cell non-autonomous roles of *rab-28* in amphid neurons could be related to a ciliary extracellular vesicle (EV) pathway, and (ii) *rab-28* expression is highly enriched in EVNs based on transcriptome analysis (*Wang et al., 2015*; *Jensen et al., 2016*). There are 27 classified EVNs in *C. elegans* that include six IL2 head neurons of males and hermaphrodites and twenty one male-specific neurons (four CEM head neurons; 16 RnB tail ray neurons; one HOB tail hook neuron) (*Figure 2A*).

To confirm that *rab-28* is expressed in EVNs, we examined transgenic male animals co-expressing the EVN reporter *klp-6p*::tdTomato (*Morsci and Barr, 2011*) and a GFP reporter under the control of *rab-28* 5'UTR (promoter) sequence (*rab-28p*::sfGFP). *rab-28* is expressed in the IL2 neurons present in both males and hermaphrodites, as well as all 21 male-specific EVNs (*Figure 2A*). Next we explored RAB-28 subcellular localization in male-specific EVNs using our GFP::RAB-28$^{Q95L}$ reporter. Like amphid and phasmid channel neurons, RAB-28 is enriched at the PCM of CEM and RnB neurons (*Figure 2B*). However, unlike amphid and phasmid cells, a pool of RAB-28 also occurs within the distal ends of the CEM and ray RnB cilia (see arrowheads in *Figure 2B*). Moving RAB-28-positive IFT particles

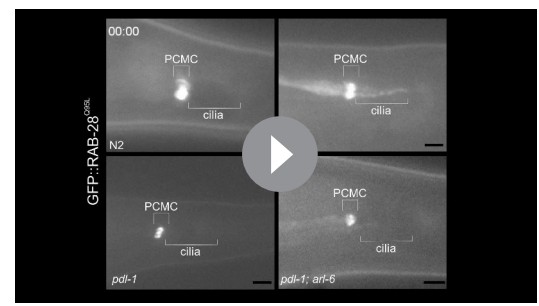

**Video 1.** Representative movies of GFP::RAB-28$^{Q95L}$ IFT behavior in the phasmid cilia of N2 (WT), *arl-6* and *pdl-1;arl-6* mutant hermaphrodites. Anterior is to the left. Movies are played at 5 frames per second. Scale bars; 2 μm. PCMC: periciliary membrane compartment.
https://elifesciences.org/articles/50580#video1

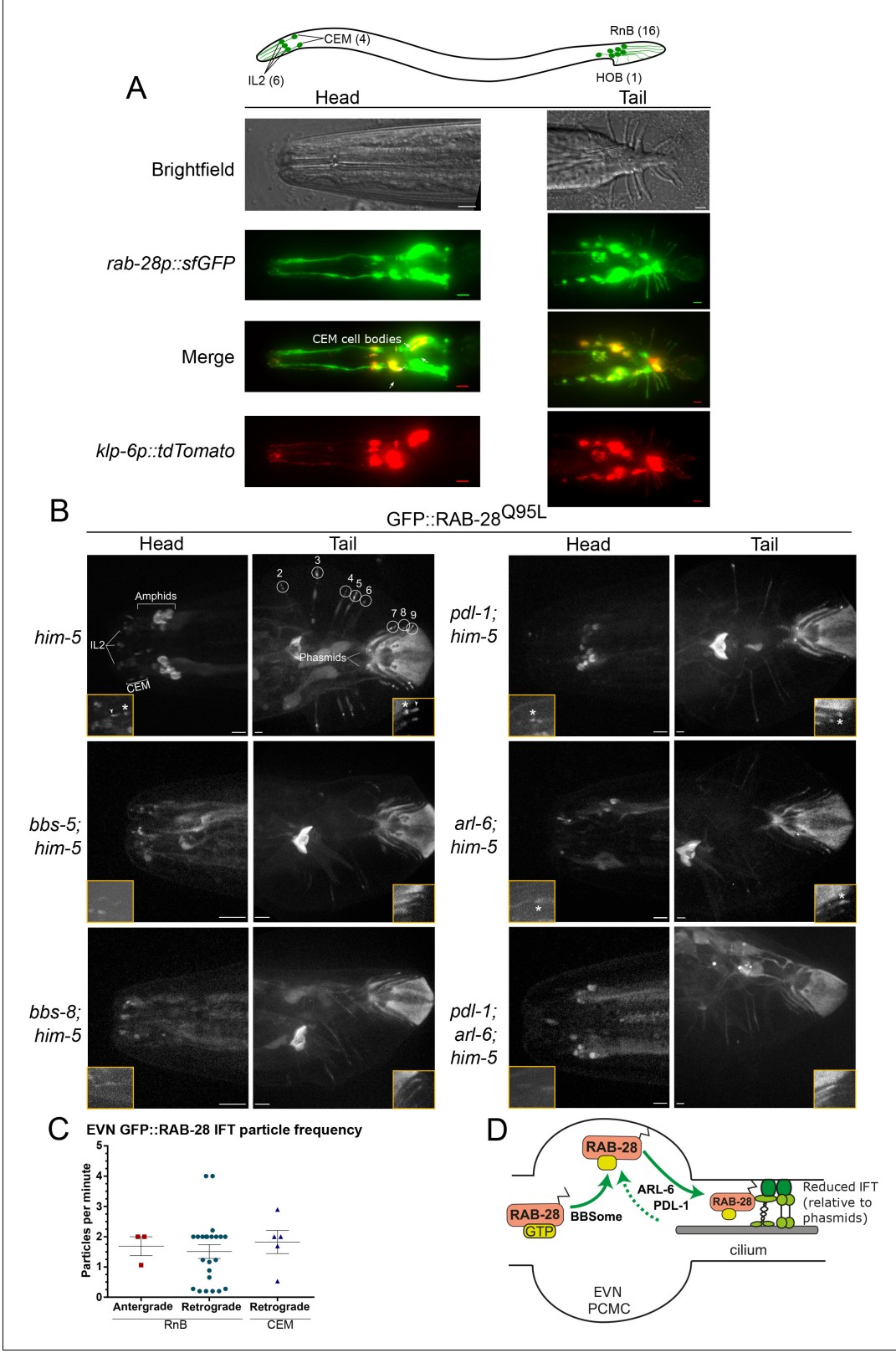

**Figure 2.** RAB-28 is expressed in and trafficked to EVN cilia via a modified BBSome-ARL-6-PDL-1 pathway. (**A**) Epifluorescence z-projections of the heads and tails of *C. elegans* males expressing *rab-28p::sfGFP* and *klp-6p:: tdTomato* (EVN cilia-specific reporter). Arrows denote the CEM neurons. Brightfield images of male head and tail are also shown for clarity. Anterior is to the left. Scale bars; 5 μm. (**B**) Representative images of the male head and *Figure 2 continued on next page*

*Figure 2 continued*

tail regions of the indicated genotypes expressing GFP::RAB-28<sup>Q95L</sup> in EVNs. Insets show higher magnification images of CEM (head) and RnB (numbered 1–9 in the tail) cilia. Asterisks indicate PCMC; white arrowheads indicate accumulated GFP::RAB-28<sup>Q95L</sup> in the distal region of CEM and RnB cilia. Anterior is to the left. Scale bars; 5 μm. (C) Scatter plots of GFP::RAB-28 IFT particle frequency in CEM and RnB male cilia. Error bars show SEM. Data are from 26 worms. (D) Schematic of proposed model for RAB-28 transport to EVN cilia. Scale bars; 10 μm. PCMC: periciliary membrane compartment.

The online version of this article includes the following source data for figure 2:

**Source data 1.** Data for *Figure 2C*.

were rarely detected in CEM and RnB cilia (*Figure 2C* and *Video 2*), consistent with our previous report showing that IFT in CEM cilia is less frequent than in amphid cilia (*Morsci and Barr, 2011*). Interestingly, almost all detectable RAB-28-positive IFT trains in male EVN cilia move in the retrograde direction (*Figure 2C*). We did not observe RAB-28 in environmentally-released EVs, indicating that RAB-28 is not an EV cargo.

To investigate the RAB-28 ciliary targeting pathway in EVNs, we expressed the GFP::RAB-28<sup>Q95L</sup> reporter in *him-5(e1490)*-containing (induces high incidence of male progeny) mutants of the BBSome (*bbs-5, bbs-8*), *arl-6* and *pdl-1*. In EVN cilia, GFP::RAB-28<sup>Q95L</sup> PCM localization was absent in *bbs-5(gk507)* and *bbs-8(nx77)* males, whilst reduced in *arl-6(ok3472)* and *pdl-1(gk157)* males (*Figure 2B*), similar to phenotypes observed in the amphid and phasmid channel neuronal cilia (*Figure 2B*). Also, the distal pool of RAB-28 in CEM and RnB cilia was reduced or absent in all of these mutants (*Figure 2B*). Like in amphid and phasmid neurons, IFT movement of RAB-28<sup>Q95L</sup> was also absent in *bbs-5(gk507), bbs-8(nx77)* and *pdl-1(gk157)* mutant EVN cilia. Surprisingly, RAB-28<sup>Q95L</sup> IFT movement was also not detectable in the EVN cilia of *arl-6(ok3472)* males, which is in striking contrast to the increased frequency of RAB-28-positive IFT trains in amphid and phasmid neurons (*Figure 1—figure supplement 1C*). Analysis of *pdl-1(gk157);arl-6(ok3472)* double mutant EVNs revealed that GFP::RAB-28<sup>Q95L</sup> is diffusely localized in the cytoplasm, with no PCM enrichment (*Figure 2B*). Thus, *pdl-1* and *arl-6* demonstrate an additive genetic relationship in targeting RAB-28 to EVN cilia, which is in contrast to that of amphid and phasmid neurons where the relationship is suppressive.

Together, these data reveal that the BBSome/ARL-6/PDL-1 network that regulates RAB-28 localization and IFT behavior in amphid and phasmid neurons is also functioning in EVNs. Although similar, there are differences in how the network is deployed in these different cell types. In amphid and phasmid neurons, GTP-bound RAB-28 is enriched at the PCM and undergoes continuous IFT, which is promoted by PDL-1 and inhibited by ARL-6 (summarized in *Figure 1* model). In EVNs, the PCM is also a site of RAB-28<sup>Q95L</sup> enrichment along with an additional RAB-28<sup>Q95L</sup> pool in the distal cilium. However, RAB-28 IFT events are infrequent in this cell type and positively regulated by ARL-6 (summarized in *Figure 2D* model).

## *C. elegans* BBS gene mutants phenocopy the amphid sensory organ defects of *rab-28* mutants

The identification of ARL-6-BBSome-PDL-1 ciliary targeting pathways for RAB-28 prompted us to investigate possible functional relationships between RAB-28 and its trafficking machinery. The amphid sensory organ (or sensillum) contains 10 rod-like cilia (from 8 neurons) located in an environmentally-exposed pore, formed by the dendrites and ciliary axonemes punching through surrounding amphid sheath and socket glial cell processes (*Ward et al., 1975*; *Doroquez et al., 2014*). Previously we showed

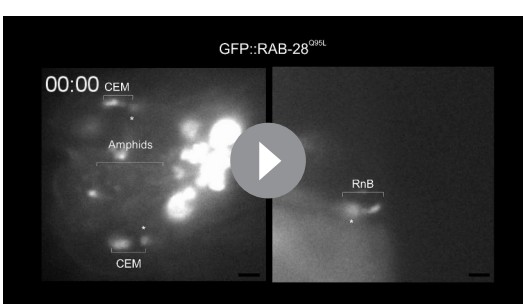

**Video 2.** Representative movies of GFP-RAB-28<sup>Q95L</sup> IFT behavior in CEM and RnB cilia of *him-5(e1490)* males. A RAB-28-positive IFT particle can be seen in the bottom CEM and RnB cilia. Higher frequency IFT movement of RAB-28 is evident from the amphid cilia. PCMCs are marked by asterisks. Movies are played at 6 fps. Anterior is to the left. Scale bars; 2 μm.

https://elifesciences.org/articles/50580#video2

that RAB-28 disruption via overexpression of GTP or GDP-preferring RAB-28 variants causes cell non-autonomous defects in amphid sensory compartment size and the surrounding sheath cell (*Jensen et al., 2016*). To investigate if the RAB-28 ciliary trafficking machinery also serves roles in amphid sensory compartment size and sheath regulation we performed TEM on chemically-fixed BBSome gene and *arl-6(ok3472)* mutant hermaphrodites. We found that *bbs-5(gk507)* and *bbs-8 (nx77)* mutant hermaphrodites abnormally accumulate electron dense matrix filled vesicles in the sheath cell process similar to RAB-28^T26N^-overexpressing worms, although sometimes at higher levels (*Figure 3A*). Additionally, much like hermaphrodites overexpressing RAB-28^Q95L^ (GTP-preferring) (*Jensen et al., 2016*), *bbs-5(gk507)* and *bbs-8(nx77)* mutant hermaphrodites consistently exhibit highly distended amphid compartments,2–3 times the size of WT,although with much darker staining of the extracellular matrix (ECM) (*Figure 3A,B*). Also, at the level of the ciliary middle segments and transition zones, where the compartment is most enlarged, *bbs-5(gk507)* and *bbs-8(nx77)* mutant compartments each contain 11 ciliary axonemes, instead of the usual 10 (*Figure 3B,C*). A weaker phenotype was observed in the *arl-6(ok3472)* mutant; whilst there are electron dense deposits in the sheath cell at the ciliary middle segment, transition zone and PCMC levels of the amphid sensory organ, compartment size and ECM density are normal (*Figure 3A*).

Together, these data show that disruption of the BBSome, or ARL-6 to a lesser extent, phenocopies the amphid sensory organ and sheath cell defects observed in *rab-28*-disrupted worms (overexpressing RAB-28^T26N^ or RAB-28^Q95L^). As with *rab-28*, *C. elegans* BBS genes are expressed exclusively in ciliated neurons (*Ansley et al., 2003*; *Fan et al., 2004*), indicating that their effects on the amphid sheath cell are also likely due to cell non-autonomous functions. Thus, in the amphids, the BBSome, ARL-6 and RAB-28 function together in a common cell non-autonomous pathway that regulates sensory organ size.

## RAB-28, BBS-8 and ARL-6 negatively regulate EV shedding

To directly investigate a role for RAB-28 and its transport machinery in EV regulation, we first examined the localization of fluorescent protein-tagged EV cargo in *rab-28* mutants. We used two deletion alleles of *rab-28*, the previously studied *gk1040* null allele (998 bp deletion) (*Jensen et al., 2016*) and the *tm2636* allele, which is a 147 bp deletion that is expected to affect nucleotide binding and thus, disrupt RAB-28 protein function (*Figure 4A* and *Figure 4—figure supplement 1*). PKD-2 is a TRP polycystin-2 expressed in male-specific EVNs, including the CEM neurons, and localizes to cilia and ciliary EVs (*Barr et al., 2001*; *Wang et al., 2014a*). In control males expressing a rescuing PKD-2::GFP transgene, PKD-2 is observed at the PCMC and ciliary tip of CEM neurons, and within EVs that are shed into the cephalic sensory organ lumen and environmentally released outside of the male (*Figure 4A–C*). Note that we previously showed that PKD-2::GFP localization is indistinguishable from endogenous PKD-2 localization measured by anti-PKD-2 antibody staining (*Barr et al., 2001*; *Maguire et al., 2015*). In *rab-28(tm2636)* mutant males, total PKD-2::GFP levels at the CEM ciliary region from the PCMC to the tip are similar to control (*Figure 4D*). However, fluorescence intensity measurement analyses revealed a subtle yet consistent change in the pattern of PKD-2::GFP distribution in *rab-28(tm2636)* males indicating a ciliary localization defective (Cil) phenotype. Specifically, unlike control animals where PKD-2::GFP is most intense at the PCMC region of the CEM cilium, *rab-28(tm2636)* mutants display abnormally high PKD-2::GFP levels at more distal parts of the CEM cilium region (*Figure 4B,C*).

Cil defects could arise due to defects in PKD-2 trafficking to cilia or in PKD-2-labeled EV biogenesis, cilia base EV shedding into lumens, or ciliary EV release into the surrounding media (*Wang et al., 2014a*; *Maguire et al., 2015*; *Silva et al., 2017*; *O'Hagan et al., 2017*; *Bae et al., 2006*). Since the abnormal PKD-2::GFP accumulation at more distal regions of the *rab-28(tm2636)* cephalic sensory organ appears to extend beyond that predicted by sole localization to the relatively narrow CEM cilium, we suspected that this excess GFP signal derives from abnormally high levels of PKD-2::GFP-labeled EVs in the cephalic lumen. To determine if *rab-28* regulates environmental EV release similar to IFT and tubulin code regulators (*Wang et al., 2014a*; *Maguire et al., 2015*; *Silva et al., 2017*; *O'Hagan et al., 2017*; *Bae et al., 2006*), we compared the number of PKD-2::GFP-labeled EVs in mounting media surrounding control and *rab-28(tm2636)* males. We found that control and *rab-28(tm2636)* males release similar numbers of PKD-2::GFP-labeled EVs, ruling out a function for RAB-28 in environmental EV release of PKD-2 and suggesting a possible defect whereby EV shedding into the sensory organ lumen is abnormally upregulated (*Figure 4E*).

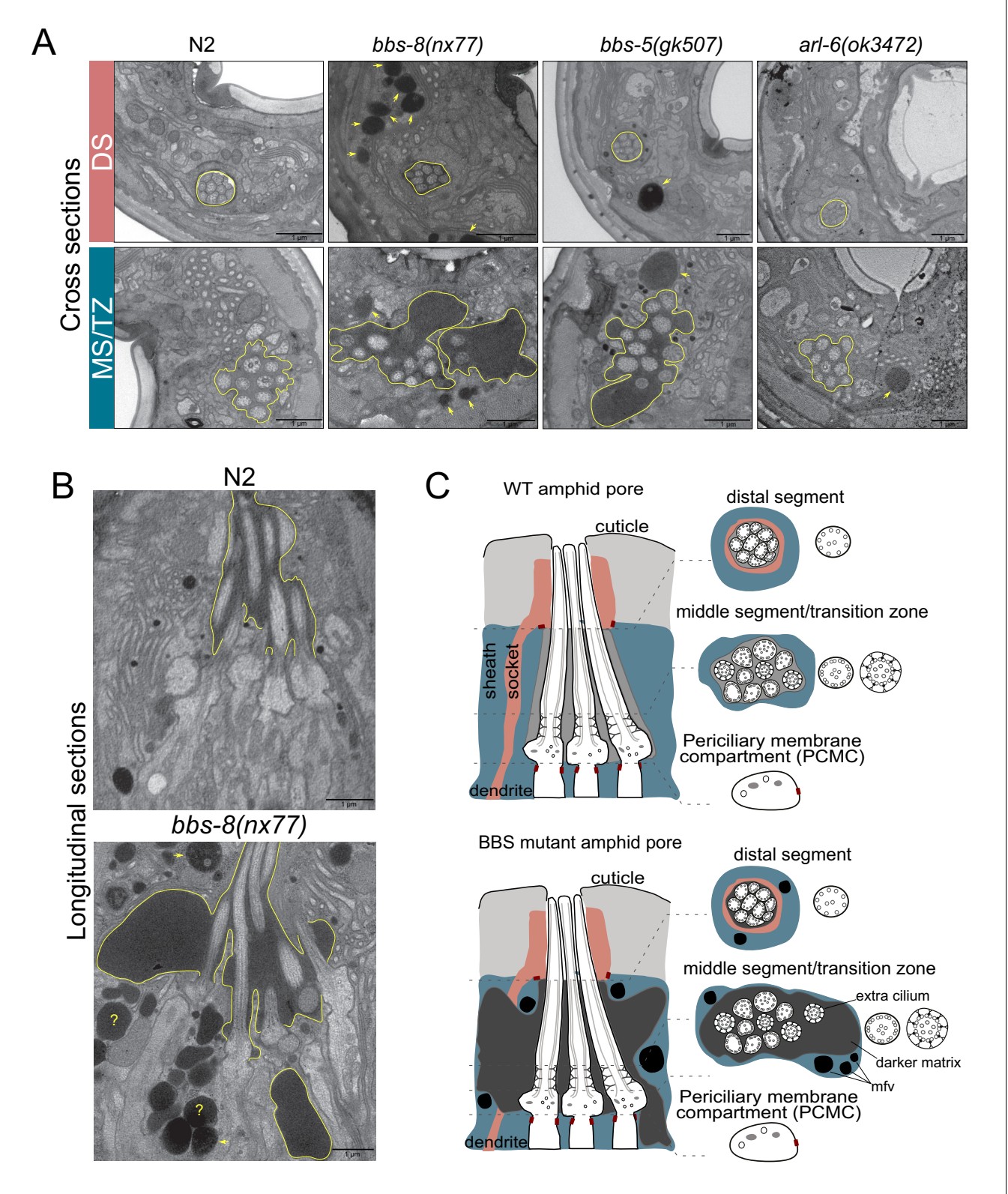

**Figure 3.** BBSome and *arl-6* mutant hermaphrodites display defects in amphid sensory organ structure and/or associated glia. (**A**) Transmission electron microscopy (TEM) images showing the amphid channels of the indicated genotype in cross section, at the positions of ciliary distal segments (DS) and middle segments (MS)/transition zones (TZ). Arrows point to matrix-filled vesicles (mfv) within the cytoplasm of the sheath glial cell that surrounds the amphid cilia. The extracellular matrix-filled amphid compartment volume is also highlighted. Scale bars; 1 µm (all panels). n = 4 hermaphrodites for WT,

*Figure 3 continued on next page*

*Figure 3 continued*

n = 1 hermaphrodite for *arl-6*, *bbs-5* and *bbs-8* (two amphid organs imaged per hermaphrodite). (**B**) TEM images of longitudinal sections through the amphid cilia of WT and *bbs-8* mutant worms, highlighting the expanded compartment volume and accumulated mfv in the sheath cell (arrows). N = 2 hermaphrodites imaged per genotype, with two amphid organs imaged per hermaphrodite. Question marks denote densities for which identification as either an mfv or an expanded pore region is ambiguous. Scale bars; 1 μm. (**C**) Cartoon representations of amphid organ ultrastructure in WT and BBSome mutant hermaphrodites, in longitudinal and cross section. Only three cilia are shown in the longitudinal schematics for simplicity.

To further explore a role for *rab-28* in ciliary EV biogenesis and shedding, we examined the localization pattern of CIL-7, a peripheral membrane protein that localizes to cilia and ciliary EVs in all 27 EVNs (*Maguire et al., 2015*). In control males expressing a rescuing CIL-7::GFP transgene, CIL-7::GFP localizes to the dendrite, PCMC and cilium of the CEM neuron, and is found in EVs released into the environment. In contrast to the relatively subtle PKD-2::GFP Cil phenotype, the *rab-28 (tm2636)* cephalic sensory organ abnormally accumulates large amounts of CIL-7::GFP in ciliary regions (*Figure 4F–H*). A similar strong Cil phenotype was also observed in *rab-28(gk1040)* worms (*Figure 4—figure supplement 2*). Wild-type and *rab-28(tm2636)* males release similar numbers of CIL-7::GFP-containing EVs in mounting media (*Figure 4I* and *Figure 4—figure supplement 2C*). These results indicate a role for RAB-28 in regulating the abundance and distribution of ciliary proteins expressed in the EV-releasing CEM neurons. Our data also indicate that RAB-28 is not required for the release of PKD-2- and CIL-7-positive EVs into the environment.

Finally, we examined the localization of CIL-7::GFP in RAB-28 transport regulator mutants. Similar to *rab-28* mutants, we found Cil defects in *bbs-8(nx77)* and, to a lesser extent, *arl-6(ok3472)* mutants, with the ciliary EV marker CIL-7::GFP accumulating at the PCMC and around the axonemal region of CEM cilia (*Figure 4J–L*). This observation is consistent with a similar previously reported PKD-2::GFP Cil phenotype in a *bbs-7* mutant (*Bae et al., 2008*; *Braunreiter et al., 2014*). In contrast, *pdl-1 (gk157)* mutants show no gross CIL-7 localization defects (*Figure 4J,L*). Similar to *rab-28* mutant males, we found no defect in the number of CIL-7-labeled EVs released into the media by *bbs-8 (nx77)*, *arl-6(ok372)* and *pdl-1(gk157)* mutant males (*Figure 4M,N*).

Together, these data indicate that RAB-28, BBS-8 and ARL-6, but not PDL-1, regulate the localization and levels of EV markers in the ciliated sensory organs. However, they do not regulate the release of EVs from the sensory organs into the external environment of the worm.

## RAB-28 and BBS-8 negatively regulate EV ciliary base shedding and control EV characteristics in the cephalic sensory organ

The accumulation of EV cargo markers in the ciliary region of *rab-28* and BBSome gene mutants suggests defects in ciliary EV biogenesis and shedding into the lumens of sensory organs. To investigate this possibility directly, we examined the ultrastructure of the four cephalic sensory organs in males cryofixed via high pressure freezing (HPF) to optimally preserve EVs. Each organ contains a single cilium from the CEM and CEP neurons and is environmentally exposed via a cuticular pore formed by the surrounding cephalic glial sheath and socket cells (*Figure 5A*). In wild-type males, EVs are observed in the extracellular space of the cephalic sensory organ (ie. the lumen) (*Figure 5B,C* and *Video 3*). These EVs are typically of low electron density and found predominantly in the proximal regions of the lumen, although a few occur in distal regions (*Figure 5B*; *Wang et al., 2014a*; *Silva et al., 2017*).

In *rab-28(tm2636)* male worms, the distal part of the cephalic lumen is enlarged and accumulates an excess of EVs. When compared with control counterparts, these *rab-28* mutant EVs are smaller and more uniform in size, and sometimes more electron dense, possibly indicating a change in composition (*Figure 5B*, *Figure 5—figure supplement 1B*, *Video 3*). In proximal parts of the lumen, at the level of the CEP and CEM ciliary transition zones and PCMCs, we also observed increased incidence of electron dense EVs (*Figure 5C* and *Figure 5—figure supplement 1C,E,F* and *Video 3*). Interestingly, we noticed a consistent increase in EVs in the lumens surrounding the TZs of two of the four cephalic sensory organs. There was no bias for EV accumulation in any one of the four specific organs (ie. left/right, dorsal/ventral). Thus, RAB-28 loss disrupts the number and nature of EVs in the lumens of cephalic sensory organs. Notably, electron tomographic analysis revealed that the cephalic sensory organ and CEM ciliary tip in the *rab-28(tm2636)* mutant retains normal exposure to the environment, indicating that pore blockage or a ciliogenesis defect is not the cause of EV

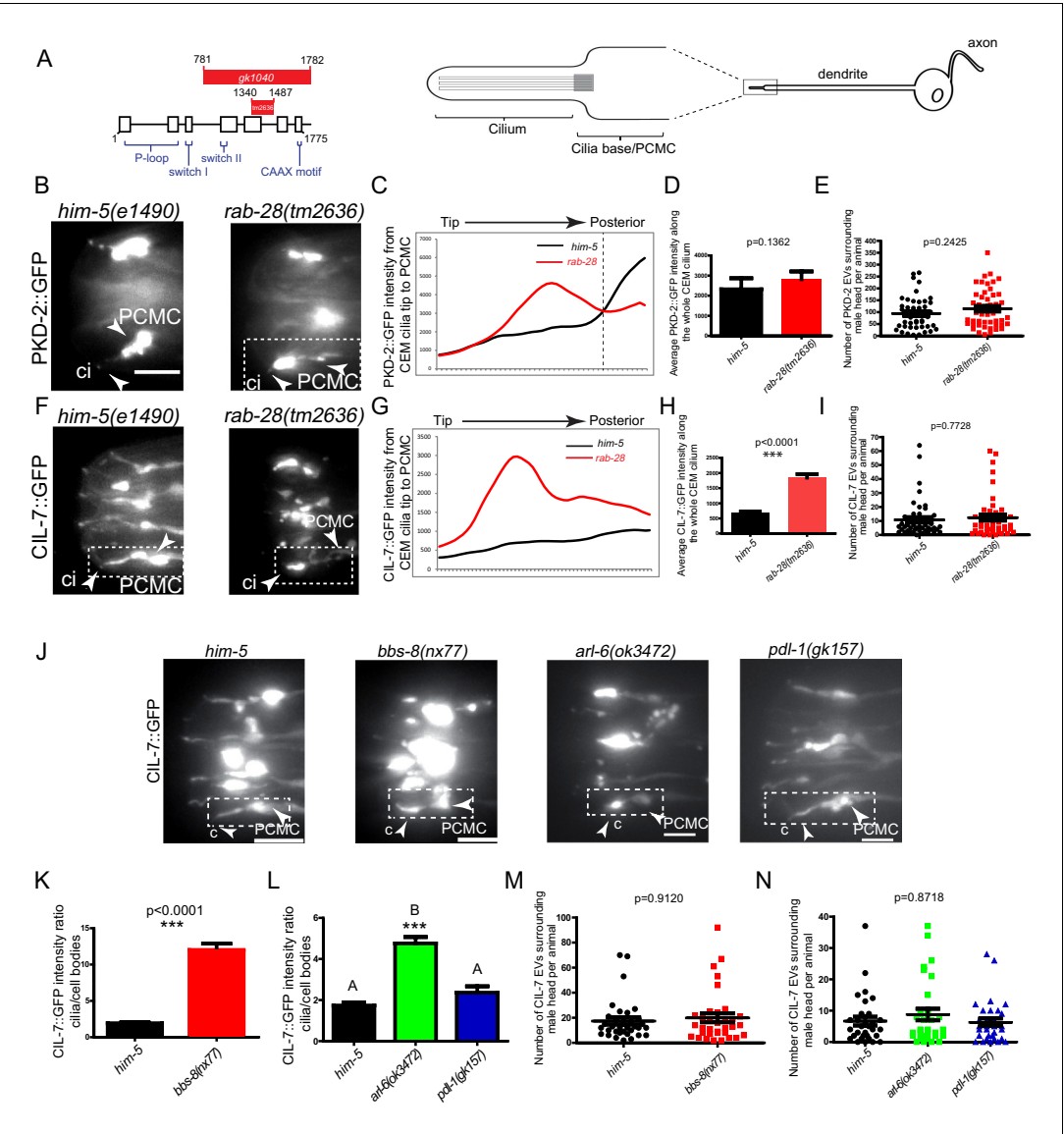

**Figure 4.** RAB-28 and BBSome components regulate the localization of ciliary EV cargoes in EV-releasing CEM cilia. (A) Left- schematic showing the location of the deleted regions in the *gk1040* and *tm2636* mutant alleles of *rab-28.* Right- cartoon modified from *Bae et al. (2006)* depicting the morphology of the CEM neurons of *C. elegans.* PCMC; periciliary membrane compartment. (B, F) Fluorescence micrographs of CEM cilia of control and *rab-28(tm2636)* males expressing PKD-2::GFP or CIL-7::GFP. Dotted white boxes mark one CEM cilium (ci), including the PCMC (periciliary membrane compartment) at the base. Scale bars; 5 µm. (C, G) Plot profiles of PKD-2::GFP and CIL-7::GFP intensity across different points along the cilium in control and *rab-28(tm2636)* adult males. Traces run from the ciliary tip and posterior towards the PCMC. Each data point represents the average GFP intensity at an individual point on several cilia in many animals of each genotype. *rab-28(tm2636)* males (n = 57 cilia from 36 males) accumulate more PKD-2 anterior to the site where PKD-2 accumulation is greatest in control males (n = 46 cilia from 32 males). *rab-28(tm2636)* males (n = 32 cilia from 18 males) accumulate more CIL-7 along the length of the cilium compared to controls (n = 48 cilia from 34 males). (D, H) Bar charts depicting mean PKD-2::GFP and CIL-7::GFP intensity along the cilium length of control and *rab-28(tm2636)* adult males. Error bars depict SEM. p values calculated by a Mann-Whitney test. For control, n = 40 (D) and 47 (H); for the *rab-28* mutant, n = 50 (D) and n = 32 (H) cilia. Data is from three separate experiments. (E, I) Scatter plots depicting the number of PKD-2::GFP- and CIL-7::GFP-positive EVs surrounding the male head per animal between control and *rab-28 (tm2636).* Horizontal line depicts the mean. Error bars depict SEM. p values calculated by a Mann-Whitney test. For control, n = 47 (E) and n = 46 (I) males; for the *rab-28* mutant, n = 48 (E) and n = 47 (I) males. (J) Fluorescence images of CIL-7::GFP in the male heads of the indicated genotypes. Scale bars; 5 µm. (K, L) Bar charts depicting

*Figure 4 continued on next page*

*Figure 4 continued*

the ratio of CIL-7::GFP intensity between the ciliary and cell body regions in the indicated genotypes. Error bars show SEM. p values in K determined by Mann-Whitney test. n = 21 males for both genotypes in K and n = 27 males for all genotypes in L. Letters above each dataset in L indicate results of statistical analysis; data sets that do not share a common letter are significantly different at p<0.0005 (Kruskal–Wallis test with Dunn's post-hoc correction). (**M, N**) Scatter plots depicting the number of CIL-7::GFP-labeled EVs released from the indicated genotypes. Error bars depict SEM. p values determined by a Mann-Whitney test (M) or Kruskal-Wallis test with Dunn's multiple comparisons (N). n = 34 (M) and n = 31 (N) males.

The online version of this article includes the following source data and figure supplement(s) for figure 4:

**Source data 1.** Data for 4C-E, 4G-I, 4 K-N.
**Figure supplement 1.** *tm2636* allele of *rab-28*.
**Figure supplement 2.** *rab-28(gk1040)* phenocopies the CIL-7 mislocalization phenotype of *rab-28(tm2636)* worms.
**Figure supplement 2—source data 1.** Data for *Figure 4—figure supplement 2B,C*.

accumulation in the distal lumen (*Figure 5—figure supplement 1D*). Also, *rab-28(tm2636)* worms retain the normal CEM ciliary axonemal AB tubule split that is important to EV release (*Silva et al., 2017*; *O'Hagan et al., 2017*). We conclude from these data that *rab-28(tm2636)* mutant males produce and shed excessive amounts of abnormally stained and sized EVs into the cephalic sensory organ. These data suggest that RAB-28 regulates EV cargo sorting and production (biogenesis) and that RAB-28 acts as a negative regulator of EV shedding without affecting environmental EV release. Our findings also show that RAB-28 negatively regulates cephalic sensory compartment size, in agreement with what we previously reported for RAB-28 in the amphid organs (*Jensen et al., 2016*).

In *bbs-8(nx77)* male worms, the cephalic lumen is distended and filled throughout (ie, proximal and distal regions) with abnormally large numbers of EVs (*Figure 5D*). Compared with *rab-28 (tm2636)* worms, the accumulated EVs in *bbs-8(nx77)* mutant lumens are more numerous, and more varied in size. We also observed electron dense matrix filled vesicles within seventy five percent of *bbs-8(nx77)* mutant cephalic sheath glia (*Figure 5D*), matching the amphid sheath cell phenotype in these worms (*Figure 3*). Similar to *rab-28(tm2636)* mutants, the ultrastructure of *bbs-8(nx77)* mutant CEM cilia is grossly normal. Thus, like RAB-28, the BBSome also negatively regulates EV shedding in the cephalic organ and the size of the organ's lumen. The differences in the severity of the EV phenotype in *bbs-8(nx77)* and *rab-28(tm2636)* mutants, and in the appearance of EVs that accumulate in both these worms, indicate a greater role for BBS-8 in EV regulation, possibly because the BBSome regulates the trafficking of multiple EV regulators and not just RAB-28.

## BBS-8 but not RAB-28 suppresses EV shedding into the amphid sensory organ

While analyzing EVs in the cephalic sensory organs of *rab-28* and *bbs-8* mutants, we were surprised to observe ectopic EVs in the amphid sensory organ lumens of cryofixed male *bbs-8* but not *rab-28* mutants. We previously observed rare EVs in the region surrounding the distal-most segment of amphid channel cilia in males and have not observed release of FP-tagged EVs from amphid cilia of males or hermaphrodites (*Wang et al., 2014a*). No EVs were observed in high-resolution three-dimensional reconstruction of the hermaphrodite amphid sensory organ (*Doroquez et al., 2014*). Although we previously investigated amphid sensory organ ultrastructure in *rab-28(gk1040)* and *bbs-8(nx77)* mutant hermaphrodites (*Jensen et al., 2016* and *Figure 3*), these experiments used chemically fixed samples, which may not allow ready visualization and assessment of EVs.

As reported previously, few if any EVs are detectable in the amphid sensory organ lumen of control male animals (*Figure 6A*). A similar scenario occurs in *rab-28(tm2636)* male worms, although irregular shaped vesicular structures are occasionally observed (*Figure 6A*). In contrast, large accumulations of EVs are observed in the amphid lumen of male *bbs-8(nx77)* mutants (*Figure 6A*). These EVs are heterogeneous in size and appearance, and occur throughout the amphid sensory organ (proximal and distal regions) (*Figure 6B*). We conclude that BBS-8 suppresses abnormal amphid EV shedding in a RAB-28-independent manner.

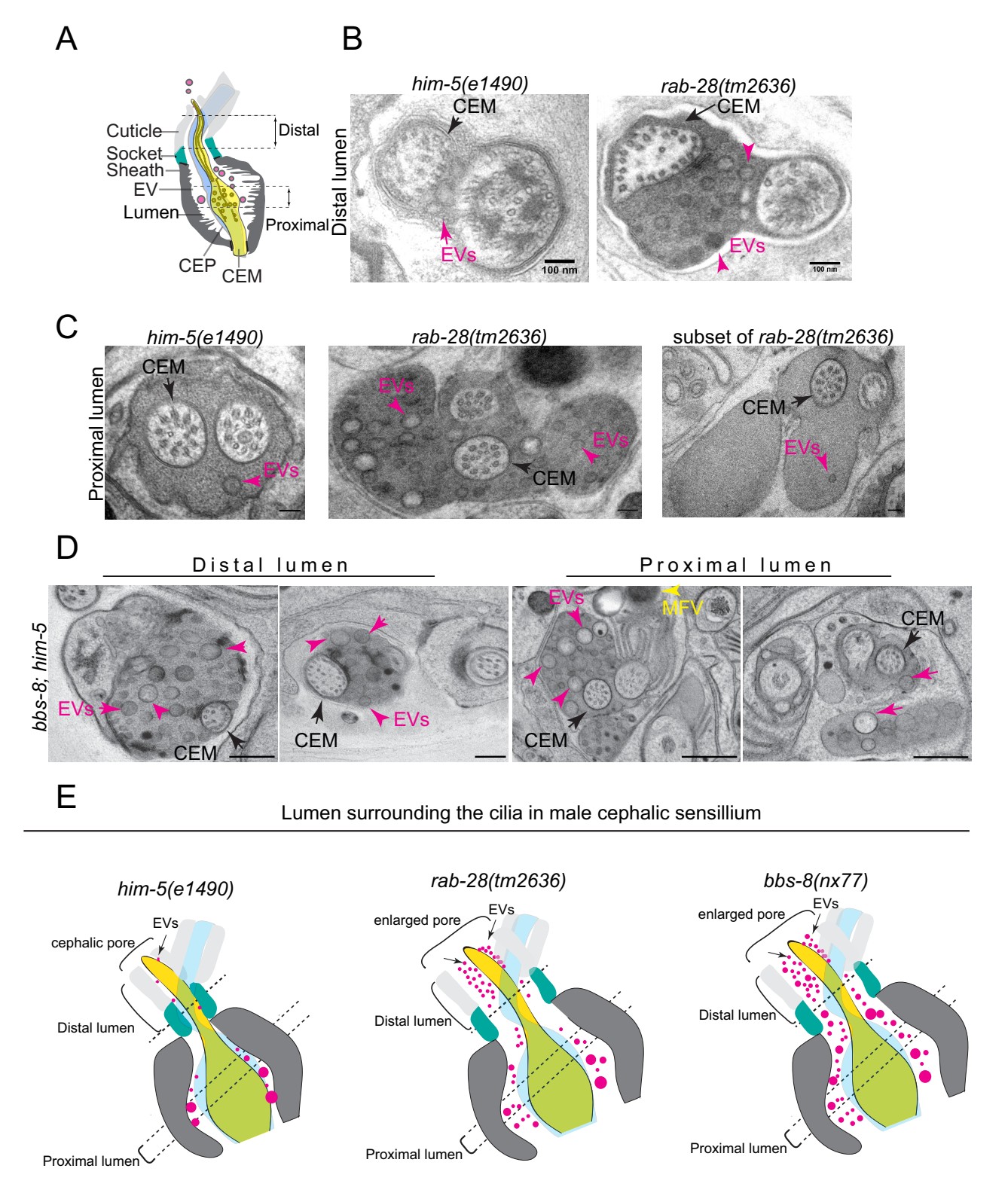

**Figure 5.** RAB-28 and BBS-8 are negative regulators of EV shedding. (**A**) Cartoon of the ultrastructure of the cephalic sensory organ reproduced from *Wang et al. (2014a)*. EVs (magenta spheres) are 'shed' from the PCMC/ciliary base into the lumen and 'released' into the environment outside of the worm. (**B**) Transmission electron micrographs of the cephalic lumen surrounding the distal region of CEM cilia. Black arrows point to the CEM and magenta arrows to EVs. *rab-28(tm2636)* accumulate significantly more EVs in the lumen distal to the singlet region of CEM compared to control males.

*Figure 5 continued*

Scale bars; 100 nm (**C**) Transmission electron micrographs of the cephalic organ at the level of the CEM cilium transition zone. Scale bars; 100 nm. A subset of the *rab-28(tm2636)* animals accumulate EVs in the cephalic lumen surrounding the TZ. (**D**) TEM cross sections of the proximal and distal regions for the cephalic lumen of *bbs-8* mutant males. Black arrows point to the CEM cilium. Ectopic EVs (magenta arrows) are observed at distal and proximal regions of the lumen. Matrix filled vesicles (MFVs) in cephalic sheath are marked by yellow arrowheads. Scale bar; 200 nm. (**E**) Cartoon depictions of the lumen surrounding the cilia in the male cephalic sensillum in control, *rab-28(tm2636)*, and *bbs-8* mutants. Color scheme is the same as the cartoon in (**A**). Brackets enclose the cephalic sensory organ pore region. *rab-28* mutant males accumulate an excess of EVs (labeled by magenta spheres and pointed to by arrows) in the lumen surrounding the more distal portion of the CEM axoneme whereas *bbs-8* mutant males accumulate excessive EVs at all levels of the cephalic lumen. *rab-28* and *bbs-8* mutants also have an enlarged cephalic pore/opening of the sensory organ.

The online version of this article includes the following source data and figure supplement(s) for figure 5:

**Figure supplement 1.** *rab-28(tm2636)* negatively regulates extracellular vesicle numbers in cephalic lumens.

**Figure supplement 1—source data 1.** Data for *Figure 5—figure supplement 1B,C,F*.

## Discussion

### The BBSome, ARL-6 and PDE6D regulate the ciliary targeting of RAB-28

We present here a network of ciliary trafficking pathways that cooperate to regulate *C. elegans* RAB-28 levels both at the periciliary membrane and within cilia. We show that BBS-5 promotes RAB-28 PCM and IFT association, thereby solidifying the requirement of the BBSome for regulating RAB-28 localization originally described (*Jensen et al., 2016*). The ortholog of the mammalian BBSome regulator, ARL-6, also maintains normal RAB-28 PCM levels, albeit not to the same extent as the BBSome. Furthermore, our results show that RAB-28 loading onto IFT trains is negatively regulated by ARL-6 and positively regulated by the orthologs of the PDE6d lipidated protein shuttle (PDL-1) and the ARL3 cargo release factor (ARL-3). Notably, *bbs-8* is epistatic (masking) to *pdl-1* for RAB-28 PCM localization, suggesting that the BBSome functions upstream of PDL-1 in the ciliary RAB-28 trafficking pathway. Our transport data here are in agreement with previous observations that the BBSome does not depend on RAB-28 for localization to phasmid cilia (*Jensen et al., 2016*). Taken together, our data suggest that the PCM is the primary membrane at which RAB-28 functions and that its levels at the PCM and ciliary membranes membrane are regulated by a combination of the BBSome, IFT, and lipidated protein transport machinery in a cell-specific manner. This latter observation is in concordance with, and expands upon, previous reports that Rab28 localization to the outer segments of mouse photoreceptors is disrupted in *Arl3* and *Pde6d* mutants (*Hanke-Gogokhia et al., 2016*; *Ying et al., 2018*). Rab28 is one of only two proteins known to undergo both IFT and lipidated protein transport, the other being INPP5E (*Fansa et al., 2016*; *Kösling et al., 2018*). In both cases, PDE6D is required for initial ciliary import and IFT for subsequent transport within cilia and, presumably, exit. This may, therefore, be a general mechanism for the ciliary targeting and distribution of lipidated proteins. Aside from its functional roles in cilia, *C. elegans* RAB-28 could serve as a useful model for investigating the trafficking of ciliary lipidated proteins and crosstalk between IFT and lipidated protein transport.

Intriguingly, we observe cell-type specific differences in RAB-28 localization and targeting. In male EVN cilia, a distinct pool of RAB-28 is present in the distal region, while the frequency of RAB-28$^{Q95L}$ IFT events is much lower than in phasmids. Whilst the requirement for the BBSome in regulating RAB-28 PCM association and IFT behavior is identical in all analyzed ciliary subtypes, PDL-1/PDE6D is partially required for maintaining RAB-28 PCM levels in EVN cilia but not amphid/phasmid cilia. In addition, an additive relationship is observed for *pdl-1* and *arl-6*,

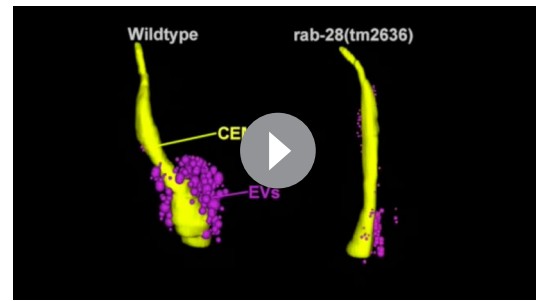

**Video 3.** Electron tomography and serial section TEM based model of the male cephalic sensory organ of control and *rab-28(tm2636)* respectively. Models depict the CEM cilium (gold), and EVs (magenta spheres). Dotted white line in movie shows position of the TZ. *rab-28* mutants ectopically accumulate excess EVs in distal regions of the cephalic lumen.
https://elifesciences.org/articles/50580#video3

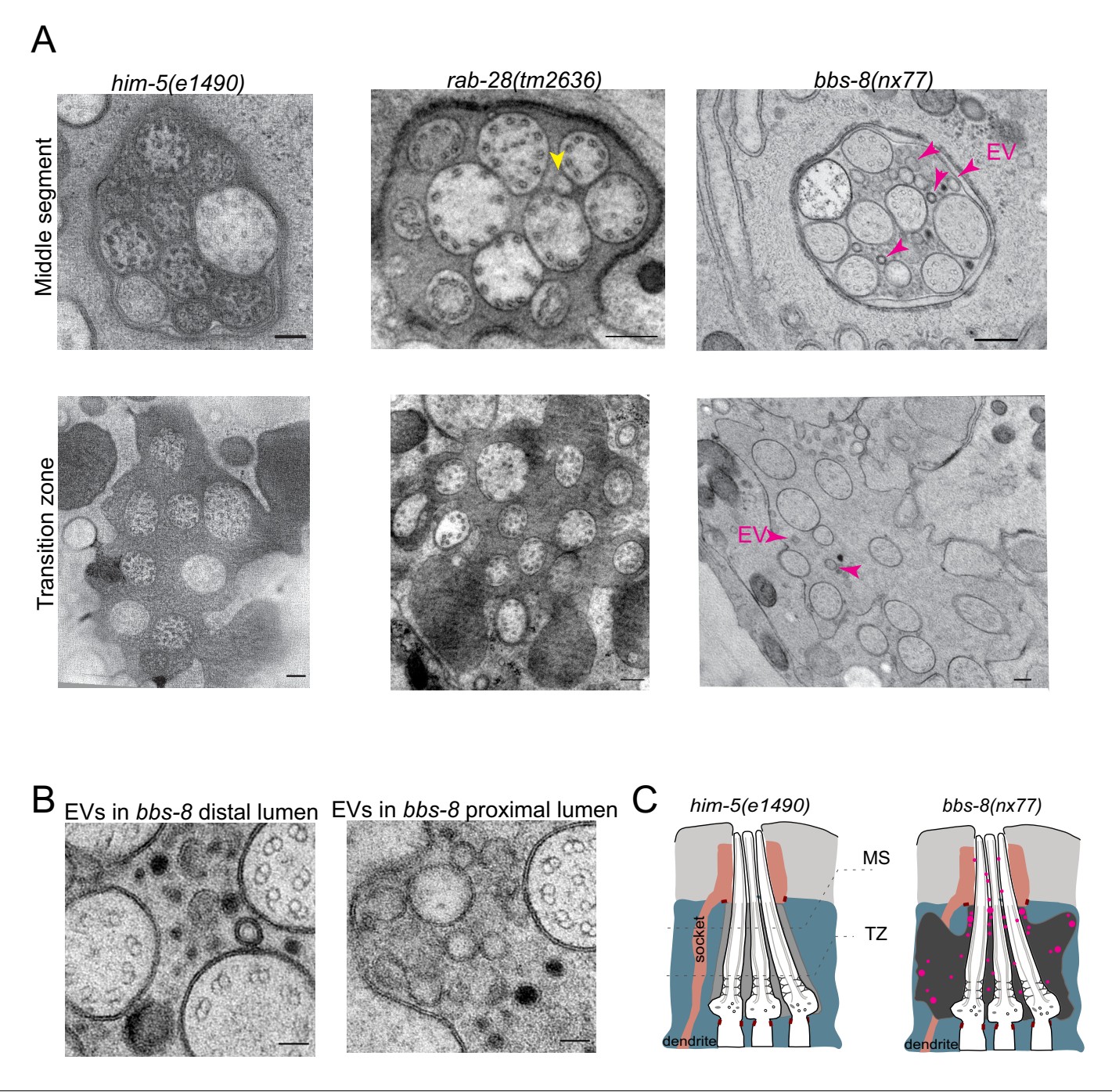

**Figure 6.** BBS-8 suppresses ectopic EV shedding in amphid sensory organs. (**A**) TEM cross sections of control, *rab-28(tm2636)*, and *bbs-8* amphid channel cilia at the middle segment (top) and at the transition zone (bottom). Yellow arrowheads in *rab-28* indicate the occasional irregular structure observed in the amphid sensory organ. Magenta arrowheads in *bbs-8* point to the ectopic EVs that accumulate in the lumen of amphid sensory organ. Scale bar; 200 nm. (**B**) High magnification images of EVs in the distal amphid channel (left) and in the proximal amphid channel (right) of *bbs-8* mutants. Scale bar; 50 nm. (**C**) Cartoon representation of EV phenotypes in control and *bbs-8* male amphid sensory organs.

The online version of this article includes the following figure supplement(s) for figure 6:

**Figure supplement 1.** *rab-28(tm2636)* male amphid sensory organs have sheath cell defects.

in that RAB-28 is fully delocalized from the PCM region of *pdl-1;arl-6* double mutant EVNs. These differences could indicate distinct relationships between PCM- and IFT-bound RAB-28 in both cell types, possibly as a result of the different RAB-28 functions in CEM versus amphid/phasmid channel cilia. Indeed, cell type-specific differences in RAB-28 ciliary trafficking correlate with distinctions in IFT and PCMC composition between amphid/phasmid and EVN cilia (*Morsci and Barr, 2011*; *Scheidel et al., 2018*).

### RAB-28 and the BBSome negatively regulate EV ciliary base shedding in EVNs

We previously proposed that ciliary RAB-28 may serve a role in EV biology (*Wang et al., 2015*; *Jensen et al., 2016*). We now confirm this hypothesis by showing that RAB-28 negatively regulates EV numbers in sensory organs containing EV-releasing neurons (EVNs). Notably, EV release from the cephalic sensory organ lumen to the animal's external environment is not affected in *rab-28* mutants, indicating that the accumulated EVs in the mutant sensory organs arise due to abnormalities in biogenesis and shedding at the ciliary base rather than environmental release. We also show that defective targeting of RAB-28 to EVN periciliary membranes, via mutation of the BBSome, elicits a similar phenotype. These findings indicate: (i) RAB-28 and the BBSome are negative regulators of EV shedding at the ciliary base, and (ii) the PCM may be the site of RAB-28's EV-related function. Taken together, our data indicate the presence of a tuneable system of EV regulation in EVNs that consists of transport regulators that control the levels and localization of EV regulators in cells and cilia. Since the prevalence, pattern, and extent of the EV phenotype is more severe in BBSome vs RAB-28 deficient worms, our data also suggests that the BBSome targets additional EV regulators, beyond RAB-28, to EVN cilia.

In *C. elegans*, very little is known about EV ciliary base shedding versus environmental release. Mutants of previously identified cilia-related EV regulators such as KLP-6, CCPP-1, TTLL-11, TBA-6, and CIL-7 showed disruption of environmental EV release with an excessive EV accumulation within sensory organs (*Wang et al., 2015*; *Maguire et al., 2015*; *Wang et al., 2014a*; *O'Hagan et al., 2017*; *Silva et al., 2017*). Our data that *rab-28* and *bbs-8* mutants are EV ciliary base shedding, but not release defective provides the very first evidence that EV shedding and release mechanisms are genetically separate. Our data is also consistent with the PCM being a major site of EV ciliary base shedding, although it cannot be ruled out that at least some of the EVs originate from other regions of the cilium or even the sheath and socket glial cells that surround the cilium (*Figure 7*).

Although it is not known why EVs accumulate in *rab-28* and BBSome gene mutants, this phenotype may arise from endosomal pathway disruption. Indeed, both Rab28 and the BBSome have previously been associated with endosomal sorting and degradative processes in Trypanosomes and *C. elegans* (*Lumb et al., 2011*; *Xu et al., 2015*; *Langousis et al., 2016*). In one model, defective ciliary protein turnover in Rab28 or BBSome-disrupted worms would lead to a compensatory mechanism whereby receptors are removed from the ciliary/periciliary membrane via EV shedding. This model is in agreement with the proposed mechanism for excess ciliary ectosome production in BBSome-depleted mammalian cells (*Nager et al., 2017*).

### The non-EVN-containing amphid sensory organ is a model for pathogenic EV shedding

Somewhat surprisingly, we also observed EV accumulations in the non-EVN-containing amphid sensory organs of *bbs-8* mutants. We consider these EVs ectopic since we almost never observe EVs in control amphids, although we cannot exclude the occurrence of EVs at specific time points during animal development. Interestingly, ectopic EVs have been described in other cilia-related contexts, associated with a pathological phenotype. For example, in the mouse retina, photoreceptor disc formation requires the suppression of EV release by the disk-specific protein peripherin (*Salinas et al., 2017*). Thus, in peripherin-disrupted mice that display photoreceptor degeneration, the ectopic EVs that surround the photoreceptor can be considered as pathological. Also, pathological EVs (ectosomes) are reported to be released from the tips of IMCD3 cells depleted of the BBSome (*Nager et al., 2017*). Thus, like the photoreceptor and IMCD3 scenarios, where EV shedding is normally rare, the amphid sensory organ represents a simple model to identify mechanisms that regulate the formation of ectopic and pathological EVs.

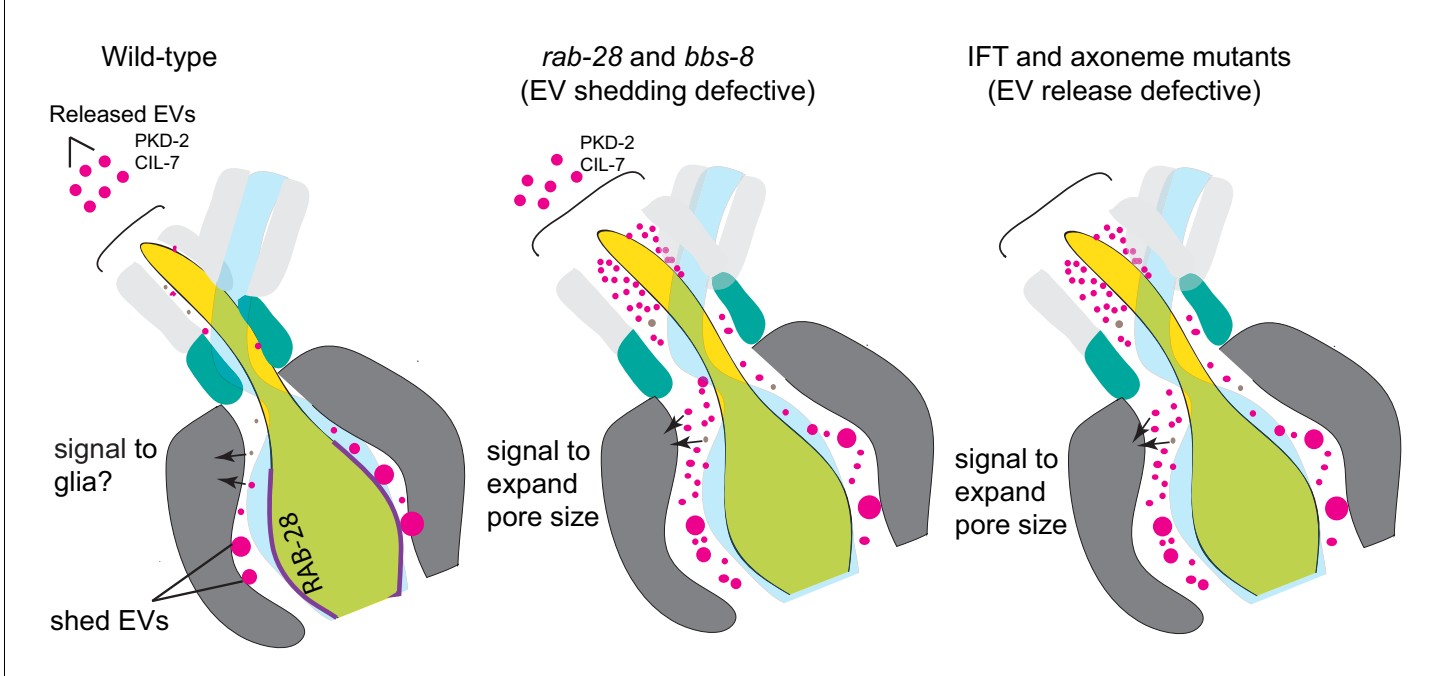

**Figure 7.** Model Cartoon depictions of the cephalic sensory organs in wild-type, EV shedding defective *rab-28* and *bbs-8* mutants, and EV release defective IFT and axoneme mutants (*ccpp-1, ttll-11, klp-6, tba-6*). Color scheme is the same as in *Figure 5*; magenta spheres represent shed EVs accumulating within sensory organs and EVs released into the environment, brown spheres represent secreted molecules within the sensory lumen. RAB-28 and BBS-8 act at the PCM to regulate EV ciliary base shedding into the lumen without affecting environmental EV release; IFT and ciliary transport components regulate EV release without abrogating EV ciliary base shedding, suggesting two sites and distinct mechanisms. Molecules released into the lumen - either ciliary EVs or secreted proteins - mediate signaling events between neurons and glia and regulate sensory organ size. EV shedding and release defective mutants both have sensory organ size defects as indicated by the expanded lumenal space in both mutant categories. RAB-28 localization in WT CEMs is indicated by purple highlighting.

## RAB-28 and the BBSome regulate sensory organ morphogenesis possibly via a role in EV-mediated neuron-glia communication

A particularly striking observation from this study is that the BBSome and RAB-28 regulate sensory organ morphogenesis in addition to EV shedding. Specifically, *bbs-8* and *rab-28* mutants display expanded sensory organ lumens and abnormally large deposits of dense vesicular material in the sheath cytoplasm that surrounds the lumen . Such deposits have been previously observed in ciliary mutants (*Perkins et al., 1986*; *Braunreiter et al., 2014*; *De Vore, 2018*) and correspond to altered forms of matrix-filled vesicles (MFVs) that normally deliver membrane and ECM to the pore (*Ward et al., 1975*; *Bacaj et al., 2008*). Indeed, the size and composition of the amphid sensory organ is thought to be critically dependent on exocytic and endocytic pathways in the sheath cell that control the amount of matrix and membrane deposited at, and retrieved from, the lumen (*Perens and Shaham, 2005*; *Singhvi and Shaham, 2019*). Additionally, signaling from amphid cilia has been proposed to regulate the localization of glial factors such as DAF-6 and LIT-1 that regulate the size of the sensory compartment/sensory organ (*Oikonomou et al., 2011*). Given these observations, we propose that RAB-28 and the BBSome coordinate sensory organ morphogenesis via signaling events that mediate neuronal-glial interactions (*Figure 7*; *Table 1*).

A tantalizing hypothesis is that RAB-28 and the BBSome fulfill this role via their EV regulatory function. Indeed, EV exchange between neurons and glia has been documented in vitro and in vivo (*Lopez-Verrilli et al., 2013*; *Frühbeis et al., 2013*; *Fröhlich et al., 2014*; *Goncalves et al., 2015*). However, whether the abnormal EV and sensory organ expansion phenotypes in *rab-28* and *bbs-8* mutants are directly linked is not entirely clear. Whilst expanded sensory organs may be due to aberrant neuronal EV-mediated signaling to glia, it could also arise from physical constraints imposed by the accumulated EVs. Alternatively, aberrant EV-independent signaling in *rab-28* and *bbs-8* mutants could lead to the downstream secretion of molecules that signal the organ to expand, with ectopic

EVs as a consequence, rather than a cause, of the enlargement. An important question to answer, therefore, is whether the contents of the EVs observed in mutant sensory organs include molecules that signal to sheath cells. Depletion of such molecules would be predicted to suppress sensory organ expansion in *rab-28* and BBSome gene mutants. Also, further work is needed to determine the origins, cargo contents, cellular targets, and functions of shed EVs to learn about the relationship between EV shedding and regulation of sensory organ compartments. Despite the gaps in our knowledge, *C. elegans* represents an excellent system for assessing the possible roles of EVs in regulating the formation of a simplified organ (ie. a sensory organ) in an intact animal.

## EV shedding regulation and the pathogenesis of RAB28 and BBSome-associated retinal dystrophy

The association of RAB28 and BBSome gene disruption with cone-rod dystrophy (*Roosing et al., 2013*; *Riveiro-Álvarez et al., 2015*; *Lee et al., 2017*; *Ying et al., 2018*) and retinitis pigmentosa (*Waters and Beales, 2011*; *Weihbrecht, 2017*) raises the possibility that EV regulatory functions contribute to human retinal disease. As mentioned above, suppression of ciliary EV release is a crucial step in the formation of the discs of the outer segments (OS) of photoreceptors (*Salinas et al., 2017*). 10% of the OS undergoes a daily renewal process that involves disc shedding and phagocytosis by neighboring retinal pigmented epithelium (RPE) cells (*Kevany and Palczewski, 2010*). However, unlike disc formation, it is not known whether OS disc shedding and RPE uptake in the mature photoreceptor is related in some way to the mechanisms governing EV shedding and EV mediated signaling. Nonetheless, our observation of EV shedding defects in *C. elegans rab-28* and *bbs-8* gene mutants, together with reduced disc shedding and/or phagocytosis reported in *Rab28* knockout mice (*Ying et al., 2018*), allows us to propose that OS disc and EV shedding processes may involve a shared Rab28 mechanism. However, the precise nature of this mechanism is unclear given that Rab28 positively regulates mouse OS shedding but negatively regulates *C. elegans* EV shedding. Furthermore, although *Bbs8* mutant mice exhibit defects in OS morphogenesis, including disorganized discs and a shorter OS (*Hsu et al., 2017*), it remains to be determined if the BBSome functions in disc shedding. Molecules regulating ciliary EV biogenesis and shedding such as those identified in *C. elegans* sensory organs may provide insight into the molecular mechanisms underpinning OS formation and/or disc shedding.

## Final conclusions and open questions

Our work adds Rab28 to a growing list of G-proteins involved in EV biogenesis and release that includes Arf6, Rab11, Rab22A, Rab27 and Rab35 (*Muralidharan-Chari et al., 2009*; *Wehman et al., 2011*; *Ostrowski et al., 2010*; *Hsu et al., 2010*; *Wang et al., 2014b*). While the majority of Rabs associated with EV biogenesis function in exosome formation and trafficking (*Blanc and Vidal, 2018*), ciliary EVs are generally thought to be ectosomes (*Wood et al., 2013*; *Nager et al., 2017*). Rab28 is only the second Rab family member shown to have a role in cilia-related EV biogenesis, the other being IFT27/Rabl4 (*Nager et al., 2017*). Neither is associated with EV generation in other contexts. Therefore, the cilium may use some unique EV-regulators, enabling the cilium to modulate its EV output independently of the rest of the plasma membrane.

An important unanswered question is whether the EVs shed into the *C. elegans* cephalic lumen and released into the environment are exosomes, ectosomes, or both. Our previous work identified that EV biogenesis, shedding and release is normal in ESCRT pathway mutants (*Wang et al., 2014b*), and multivesicular bodies are not observed in the region of the ciliary base by TEM. Hence, our genetic and electron microscopy data are consistent with these small ciliary EVs (average diameter 104.7 ± 46.7 nm) not being exosomes but ectosomes/microvesicles instead. This conclusion is also supported by our observation of omega-shaped vesicle budding from the ciliary base (*Wang et al., 2014b*; *Silva et al., 2017*).

In the non-EVN amphid lumen, the ectopic (pathological) EVs observed in *bbs-8* mutant males are poorly characterized in terms of origin, content, regulation, and function. Identifying the cargoes within will be important to determine the mode of EV biogenesis in these sensory organs. Indeed, only a small number of ciliary EV cargos are known thus far in *C. elegans* (*Wang et al., 2014a*; *Wang et al., 2015*; *Maguire et al., 2015*; *O'Hagan et al., 2017*). Given our discovery of a potential role for EVs in neuron-glia communication, future studies should prioritize the further identification

of the content of shed EVs and how it differs between various cell type and genetic contexts. Identification of EV targeting sequences, EV purification, and correlative light electron microscopy (CLEM) approaches have great potential to address this problem.

Our studies raise several interesting questions: How and why is periciliary membrane-localized *C. elegans* Rab28 important to suppress EV shedding surrounding *distal* parts of cilia? What is the mechanism by which Rab28 and the BBSome regulate EV shedding? Why are Rab28 and the BBSome not packaged into EVs despite being closely associated with the biogenesis process? How are proteins sorted and packaged into ciliary EVs? What are the effectors and GAP/GEF regulators of Rab28? How might ciliary EVs interact with surrounding cells such as glia to regulate the shape and size of sensory organ compartments? Answers to these questions will lead to a better understanding of the fundamental biology of ciliary EVs, neuron-glia signaling, and ciliopathies with relatively unexplored EV phenotypes such as polycystic kidney disease, Bardet-Biedl Syndrome, and retinal dystrophy.

## Materials and methods

### *C. elegans* strains and maintenance

All strains were cultured according to standard protocols (*Brenner, 1974*). Briefly, worms were cultured on plates of nematode growth medium (NGM), seeded with a lawn of OP50 *E. coli*. Plates were incubated at either 20℃ or 15℃ to slow development. Reporter strains were crossed into mutant backgrounds and double mutants generated by standard crossing strategies. Mutations were followed by genotyping PCR.

### Strain list

OEB803 N2; *oqEx304[gfp::rab-28$^{Q95L}$ + unc-122p::gfp]*
OEB972 *arl-6(ok3472); oqEx304[gfp::rab-28$^{Q95L}$ + unc-122p::gfp]*
OEB970 *bbs-5(gk507); oqEx304[gfp::rab-28$^{Q95L}$ + unc-122p::gfp]*
OEB805 *bbs-8(nx77); oqEx304[gfp::rab-28$^{Q95L}$ + unc-122p::gfp]*
OEB806 *pdl-1(gk157); oqEx304[gfp::rab-28$^{Q95L}$ + unc-122p::gfp]*
OEB807 *arl-3(tm1703); oqEx304[gfp::rab-28$^{Q95L}$ + unc-122p::gfp]*
OEB971 *pdl-1(gk157); bbs-8(nx77); oqEx304[gfp::rab-28$^{Q95L}$ + unc-122p::gfp]*
OEB973 *him-5(e1490); oqEx304[gfp::rab-28$^{Q95L}$ + unc-122p::gfp]*
OEB974 *arl-6(ok3472); him-5(e1490); oqEx304[gfp::rab-28$^{Q95L}$ + unc-122p::gfp]*
OEB975 *bbs-5(gk507); him-5(e1490); oqEx304[gfp::rab-28$^{Q95L}$ + unc-122p::gfp]*
OEB976 *bbs-8(nx77) him-5(e1490); oqEx304[gfp::rab-28$^{Q95L}$ + unc-122p::gfp]*
OEB977 *pdl-1(gk157); him-5(e1490); oqEx304[gfp::rab-28$^{Q95L}$ + unc-122p::gfp]*
OEB978 *pdl-1(gk157); arl-6(ok3472); him-5(e1490); oqEx304[gfp::rab-28$^{Q95L}$ + unc-122p::gfp]*
PT621 *him-5(e1490); myIs4 [PKD-2::GFP+Punc-122::GFP]*
PT2679 *him-5(e1490); myIs23[cil-7p::gCIL-7::GFP_3'UTR+ccRFP]*
PT3189 *rab-28(tm2636); him-5(e1490)*
OEB945 *arl-6(ok3472); him-5(e1490)*
OEB947 *bbs-8(nx77); him-5(e1490)*
PT2984 *rab-28(tm2636);him-5(e1490); myIs4 [PKD-2::GFP+Punc-122::GFP]*
PT3265 *rab-28(tm2636);him-5(e1490); myIs23[cil-7p::gCIL-7::GFP_3'UTR+ccRFP]*
PT3190 *pha-1(e2123); him-5(e1490); myEx905[rab-28p::sfGFP+PBX]*
PT3356 *pha-1(e2123); him-5; myIs20 [klp-6p::tdtomato] myEx905[rab-28p::sfGFP+PBX]*
OEB948 *rab-28(gk1040); him-5(e1490) myIs23*
OEB949 *arl-6(ok3472);him-5(e1490) myIs23*
OEB951 *pdl-1(gk157); him-5(e1490) myIs23*

### Fluorescence microscopy

For PKD-2::GFP, CIL-7::GFP, *rab-28p*::sfGFP and *klp-6p*::tdTomato (**Figures 2** and **4**), L4 males were isolated the previous day to provide virgin adults for imaging the next day. Males were placed on 4% agarose pads, and immobilized in 10 mM levamisole before imaging. For CIL-7::GFP imaging (except for comparison of control and *bbs-8* mutants), males were placed in levamisole for 7–8 min prior to imaging. Epifluorescence imaging was performed using an upright Zeiss Axio Imager D1m.

Images were acquired using a digital sCMOS camera (C11440-42U30, Hamamatsu Photonics). The microscope was controlled by Metamorph 7.1 to acquire Z stacks. All images were analyzed using Fiji (*Schindelin et al., 2012*).

To compare average fluorescence intensities of CIL-7::GFP and PKD-2::GFP between wild- type and mutant males, lines were traced across the CEM cilium and the plot profile function on Fiji was used to generate intensity values across several points of the cilium. These numbers were then averaged to give an intensity value for each cilium across many animals. To obtain line graphs depicting differences in the fluorescence intensity values across different points along the CEM cilium, cilia were traced from the tip to base (PCMC) and the intensity levels at different points along the cilium were obtained for several animals. The intensity of each point along the cilium was then averaged across several cilia/animals. Only points with intensity values across all examined samples were plotted.

For EV particle quantification, EVs containing PKD-2::GFP were counted from blinded images of age-matched control and mutant adult males expressing *myIs4*[PKD-2::GFP] as described previously (*Wang et al., 2014a*). EVs containing CIL-7 were counted from images of age-matched control and mutants expressing *myIs23*[CIL-7::GFP]. EV particles were quantified from Z projections using the ROI manager tool on Fiji. All data using EV cargo reporter strains was obtained from three trials performed on three separate days; control animals were imaged alongside mutants in all experiments.

Statistical analysis: Raw data was sorted and arranged using Microsoft Excel. Statistical analyses were done using GraphPad Prism V5. Standard symbols were used to depict P values (* for $p<0.05$, ** for $p<0.005$, and *** for $p<0.0005$).

For analysis of RAB-28$^{Q95L}$ localization, hermaphrodite or male worms were placed on 5% agarose pads on glass slides and immobilized with 40 mM levamisole. L4 males were isolated the previous day to provide virgin males for imaging. Epifluorescence imaging was performed on an upright Zeiss AxioImager M1 microscope with a Retiga R6 CCD detector (Teledyne QImaging). Confocal imaging was performed on an inverted Nikon Eclipse Ti microscope with a Yokogawa spinning-disc unit (Andor Revolution) and images were acquired using an iXon Life 888 EMCCD detector (Andor Technology). All image analysis was performed using Fiji. For kymography, time-lapse (multi tiff) movies of IFT along cilia were taken at 250 ms exposure and 4 fps. Kymographs were generated from multi tiff files using the KymographClear ImageJ plugin (*Mangeol et al., 2016*). For RAB-28 localization experiments, at least three biological replicates were performed for each strain meaning that different individuals were imaged on different days; control animals were imaged alongside mutants for all experiments.

## Electron microscopy

High pressure freeze fixation (HPF) and freeze substitution for TEM on CEM cilia: control CB1490 (*him-5*), PT3189 (*rab-28(tm2636); him-5*) and OEB947 (*bbs-8(nx77); him-5*) strains were collected as L4 males the day before freeze fixation to provide virgin day 1 adult males on the day of fixation. Males were subjected to high-pressure freeze fixation using a HPM10 high-pressure freezing machine (Bal-Tec, Switzerland). Males were slowly freeze substituted in 2% osmium tetroxide, 0.2% uranyl acetate and 2% water in acetone using RMC freeze substitution device (Boeckeler Instruments, Tucson, AZ, USA) (*Weimer, 2006*). Samples were infiltrated with Embed 812 resin over 3 days prior to embedding in blocks. Most males were collected in 70 nm-thick plastic serial sections collected on copper slot grids and were post-stained with 2% uranyl acetate in 70% methanol, followed by washing and incubating with aqueous lead citrate. TEM images were acquired on either a Philips CM10 transmission electron microscope operating at 80 kV or a JEOL JEM-1400 transmission electron microscope operating at 120 kV.

TEM of the amphid sensory organs of chemically fixed BBS gene hermaphrodite mutants (*Figure 3*) was performed as previously described *Sanders et al. (2015)*. Briefly, hermaphrodites were fixed overnight at 4°C in 2.5% glutaraldehyde in Sørensen's phosphate buffer (0.1 M, pH 7.4). Samples were post-fixed in 1% osmium tetroxide and dehydrated in an ascending gradient series of ethanol concentrations prior to Epon 812 resin embedding overnight. 90 nm sections were cut using a Leica EM UC6 Ultramicrotome, collected on copper grids and post-stained with 2% uranyl acetate and 3% lead citrate. Imaging was performed on a Tecnai T12 (FEI) using an accelerated voltage of 120 kV. Electron tomography was performed as described in *Silva et al. (2017)*. Slice views of tomograms on CEM ciliary tips were exported as tiff files.

To quantify amphid sensory compartment size, the area of each amphid compartment was measured at the same section depth for each genotype. The depth chosen was at the level of the middle segments/transition zones of amphid cilia, as this is the point at which compartment size is most extensive in all cases. To assess whether a mutant accumulated an excess of matrix filled vesicles in the amphid sheath cell, the number and the size of the matrix filled vesicles in the amphid sheath were noted. Wild-type animals accumulate smaller matrix filled vesicles and therefore any mutant that accumulated noticeably large vesicles was scored as defective.

All quantifications of EV numbers were done from serial section TEM images of males of WT and *rab-28(tm2636)* animals. For measurement of EV numbers in the sub-distal region of the cephalic sensory organ lumen, EVs were counted between the regions where the CEM cilia have 18 singlet MTs up to the anterior-most section where CEM and CEP cilia share the lumen. For measurement of EV numbers at the TZ level (CEM cilium) of the cephalic sensory organ, EVs were counted between the region where all the TZ microtubules were doublets up to the region where all the TZ microtubules terminated. For measurement of EV numbers at the PCMC level (CEM cilium) of the cephalic sensory organ, EV numbers were counted in the region 140 nm posterior to where the TZ

**Table 1.** Table summarizing the amphid and cephalic sensory organ phenotypes of strains used in this study.

**Amphid sensory organ TEM phenotypes**

| Mutant | Sex | Sensory compartment size | Matrix filled vesicles in sheath | EVs in pore | Reference |
|---|---|---|---|---|---|
| *rab-28 (tm2636)* | Male | Not enlarged | Yes | Few vesicles/odd shaped particles observed surrounding MS | This work |
| *rab-28 (gk1040)* | Hermaphrodite | Not enlarged | No | ND | *Jensen et al., 2016* |
| RAB-28$^{Q95L}$ | Hermaphrodite | Enlarged | No | ND | *Jensen et al., 2016* |
| RAB-28$^{T49N}$ | Hermaphrodite | Not enlarged | Yes | ND | *Jensen et al., 2016* |
| *bbs-8* | Male and hermaphrodite | Enlarged | Yes | Lots of EVs (in male) | This work |
| *arl-6* | Hermaphrodite | Not enlarged | Yes | ND | This work |

**Male cephalic sensory organ TEM phenotypes**

| Mutant | Sensory organ size | EVs in pore | Reference |
|---|---|---|---|
| *rab-28 (tm2636)* | Enlarged | Yes, especially at the distal regions | This work |
| *rab-28 (gk1040)* | ND | ND | N/A |
| RAB-28$^{Q95L}$ | ND | ND | N/A |
| RAB-28$^{T49N}$ | ND | ND | N/A |
| *bbs-8* | Enlarged | Yes, surrounding all segments | This work |
| *arl-6* | ND | ND | N/A |

**Other phenotypes**

| Phenotype | *rab-28(tm2636)* | *rab-28(gk1040)* | *arl-6* | *bbs-8* |
|---|---|---|---|---|
| PKD-2 localization in CEMs | Mislocalized | Normal (*Jensen et al., 2016*) | ND | ND |
| CIL-7 localization in CEMs | Mislocalized | Mislocalized | Mislocalized | Mislocalized |
| Environmental release of EVs from heads | Normal | Normal | Normal | Normal |

microtubules terminate. To further ensure that the same part of the sensory organ was being assessed in the above experiments, various features of the CEP and OLQ ciliary axonemes were used as landmarks.

*C. elegans* RNA preparation and RT-PCR mRNA was isolated from mixed-stage control (*him-5*) and *rab-28(tm2636); him-5* mutants. cDNA amplified from both strains using NEB Protoscript II was used as template for PCR using gene-specific primers. The amplified PCR products from both geno-types were subsequently sequenced.

## Acknowledgements

This work was funded by the National Institutes of Health (NIH) awards DK059418 and DK116606 to MMB and OD010943 to DHH, Science Foundation Ireland (SFI) principal investigator (11/PI/1037) and SFI-BBSRC (Biotechnology and Biological Sciences Research Council) partnership (16/BBSRC/3394) awards to OEB, and an Irish Research Council (IRC) Government of Ireland postgraduate award (GOIPG/2014/683) to SC, a New Jersey Commission on Spinal cord research (NJCSCR) post-doctoral fellowship (CSCR16FEL008) to JA. MS would like to thank Dr. Erik Jorgensen for funding his postdoctoral training. We thank Juan Wang for discussion and insight on ciliary EVs, Gloria Androwski and Helen Ushakov for excellent technical assistance, Barr labmates and the Rutgers *C. elegans* community for feedback and constructive criticism throughout this project, the Conway Insti-tute imaging core facility for technical support, and WormBase. We thank Leslie Gunther-Cummins and Xheni Nishku at AECOM for assistance with high pressure freeze fixation. We also thank the National BioResource Project (Tokyo Women's Medical College, Tokyo, Japan) and *Caenorhabditis* Genetics Center (CGC) for strains. The CGC is supported by the National Institutes of Health - Office of Research Infrastructure Programs (P40 OD010440).

## Additional information

### Funding

| Funder | Grant reference number | Author |
|---|---|---|
| National Institutes of Health | DK059418 | Maureen M Barr |
| National Institutes of Health | DK116606 | Maureen M Barr |
| NIH Office of the Director | OD010943 | David H Hall |
| Science Foundation Ireland | Science Foundation Ireland (SFI) principal investigator (11/PI/1037) | Oliver E Blacque |
| BBSRC | SFI-BBSRC partnership 16/BBSRC/3394 | Oliver E Blacque |
| Irish Research Council | Government of Ireland postgraduate award GOIPG/2014/683 | Stephen P Carter |
| New Jersey Commission on Spinal Cord Research | CSCR16FEL008 | Jyothi S Akella |
| Science Foundation Ireland | SFI-BBSRC partnership 16/BBSRC/3394 | Oliver E Blacque |

The funders had no role in study design, data collection and interpretation, or the decision to submit the work for publication.

### Author contributions

Jyothi S Akella, Stephen P Carter, Conceptualization, Formal analysis, Funding acquisition, Valida-tion, Investigation, Visualization, Methodology; Ken Nguyen, Formal analysis, Investigation, Method-ology; Sofia Tsiropoulou, Ailis L Moran, Fatima Rizvi, Formal analysis, Investigation; Malan Silva, Formal analysis, Investigation, Visualization, Methodology; Breandan N Kennedy, Conceptualization, Supervision, Investigation; David H Hall, Formal analysis, Supervision, Funding acquisition,

Validation, Methodology; Maureen M Barr, Conceptualization, Resources, Formal analysis, Supervision, Funding acquisition, Investigation, Visualization, Project administration; Oliver E Blacque, Conceptualization, Resources, Data curation, Formal analysis, Supervision, Funding acquisition, Investigation, Visualization, Methodology, Project administration

### Author ORCIDs
Jyothi S Akella https://orcid.org/0000-0003-4839-4910
Stephen P Carter https://orcid.org/0000-0002-0562-8783
David H Hall http://orcid.org/0000-0001-8459-9820
Maureen M Barr https://orcid.org/0000-0003-4483-2952
Oliver E Blacque https://orcid.org/0000-0003-1598-2695

### Decision letter and Author response
Decision letter https://doi.org/10.7554/eLife.50580.sa1
Author response https://doi.org/10.7554/eLife.50580.sa2

## Additional files

### Supplementary files
- Supplementary file 1. Key resources table.
- Supplementary file 2. Primers used in this study.
- Supplementary file 3. Means and SEM for all figures.
- Transparent reporting form

### Data availability
All data generated or analyzed during this study are included in the manuscript and supporting files. Source data files have been provided for figures with statistical analyses.

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
