## [Decision Letter]

**Acceptance summary:**

The editor and reviewers found your delineation of the novel in vivo regulatory mechanisms for extracellular vesicle production exciting and compelling.

**Decision letter after peer review:**

Thank you for submitting your article "A ciliary BBSome-ARL-6-PDE6D pathway trafficks RAB-28, a negative regulator of extracellular vesicle biogenesis" for consideration by *eLife*. Your article has been reviewed by Suzanne Pfeffer as the Senior Editor, a Reviewing Editor, and two reviewers. The following individuals involved in review of your submission have agreed to reveal their identity: Guangshuo Ou (Reviewer #1).

The reviewers have discussed the reviews with one another and the Reviewing Editor has drafted this decision to help you prepare a revised submission. Both reviewers think that the work is interesting and should potentially be published in *eLife*. But they also provided concrete suggestions to improve the manuscript.

Summary:

In a 2016 PLOS Genetics paper, the Blacque and Leroux labs identified Rab-28 as a gene associated with cilia function and they made the unexpected discovery that overexpression of dominant negative Rab-28 in ciliated amphid pore neurons results in non-cell autonomous phenotypes. Namely, sheath cells accumulate dark matrix-filled vesicles in strains overexpressing dominant negative Rab-28 and the amphid pore is enlarged and filled with a light matrix in strains overpexpressing consitutively active Rab-28. As formation of the amphid pore is dependent on secretion of ECM by sheath cells, it was concluded that Rab-28 expression in ciliated neurons affects the function of sheath cells.

The nature of the communication between ciliated neurons and sheath cells was left as an open question which the current paper aims to address. In the present manuscript, the authors analyze mutants of Rab-28, of the BBSome and of ARL6 and find that Rab-28 fails to accumulate in cilia and at the periciliary membrane (PCMC) in BBSome and ARL6 mutants. Characterization of the cephalic and amphid pore by fluorescence and electron microscopy indicates that *bbs* and *rab-28* mutants accumulate extracellular vesicles (EVs) in a dilated pore filled with dark matrix. Accumulation of dark vesicles within the sheath cells is also observed in these mutants. The authors conclude that the overproduction of EVs by ciliated neurons affects matrix secretion by the sheath cells.

Although the current data set is not fully demonstrative of a role of EVs in regulating the function of sheath cells, it is a valuable contribution that advances the thesis of EVs as means of intercellular communication in a physiological context. The quality of the work is generally high, the experiments are presented clearly, and the logic is well articulated.

Essential revisions:

The major caveat of the current manuscript is the correlative nature of the argument that EVs mediate communication between ciliated neurons and sheath cells. The alternative hypothesis that secreted factors produced by ciliated cells affect the support cells is neither formulated nor tested. An important experiment would be to interfere with EV production in the *bbs* or *rab-28* mutants; the author's hypothesis predicts that the sheath cell phenotypes will be suppressed. The Barr lab previously reported that EV production is drastically reduced in *klp-6* and *ift* mutants.

1) Quoting from Jensen et al., 2016, 'sensory pore structure and function appears grossly normal in the *rab-28* deletion mutant, which is likely a null allele'. The *gk1040* allele was studied in this past publication. Yet, in the current manuscript, the same *gk1040* allele and the additional allele *tm2636* (which removes a smaller portion of Rab-28 than *gk1040*) both exhibit pronounced defects at the level of the sheath cells and the amphid pore. Could the authors comment on this discrepancy in the Discussion section?

2) The authors propose that Rab-28 functions downstream of the BBSome. Have the authors tested the localization of the BBSome in Rab-28 mutants? This experiment is particularly relevant in light of the sequence similarity between Rab-28 and RabL4/IFT27 and prior findings in mammalian cells that the BBSome accumulates in cilia when *ift27* is deleted. The model that Rab-28 functions downstream of the BBSome in the regulation of EV shedding is based on the observations that Rab-28 localization to cilia and the PCMC is reduced in BBSome mutants (current manuscript and Jensen et al., 2016) while localization of the BBSome remains normal in the *rab-28(gk1040)* mutant (Jensen et al., 2016). In light of the newly found phenotypes of Rab-28 mutants in the current paper, it would be important to re-examine the localization and dynamics of the BBSome in Rab-28 mutants and in strains expressing Rab-28 variants.

3) The authors propose that the EVs that fill the amphid pore lumen in *bbs* and *rab-28* mutants originate from budding events at the periciliary membrane (microvesicle shedding) rather than from secretion of multivesicular bodies (exosome release). Have the authors looked at the distribution of CD63-GFP, a marker of ILVs and exosomes, in the *rab-28* and *bbs* mutants? This is important. Is the release of EVs found in the amphid pore dependent upon ESCRT function?

---

## [Author Response]

Essential revisions:The major caveat of the current manuscript is the correlative nature of the argument that EVs mediate communication between ciliated neurons and sheath cells. The alternative hypothesis that secreted factors produced by ciliated cells affect the support cells is neither formulated nor tested. An important experiment would be to interfere with EV production in the bbs or rab-28 mutants; the author's hypothesis predicts that the sheath cell phenotypes will be suppressed. The Barr lab previously reported that EV production is drastically reduced in klp-6 and ift mutants.

Although we find that all mutants accumulating EVs also have pore expansion phenotypes, we agree that secreted factors, either alone or in combination with the EVs, could play a role in sensory pore morphogenesis. Indeed, as suggested by the reviewer, an excellent experiment to clarify EV-pore phenotype associations would be to examine if the expanded pore lumen in *bbs-8* and *rab-28* mutants depends on genes required for EV production. However, as of yet, we have not identified such genes. Please note that our previous work identified *klp-6, ccpp-1, ttll-11, tba-6*, and *cil-7* as positive regulators of ciliary EV release into the environment, but with no loss of EV shedding abilities within the cephalic sensory organ in all these mutants (as defined by fluorescence microscopy and TEM analyses). Thus, we have not identified mutants that completely block ciliary base EV shedding, and hence we cannot do the suggested epistasis experiment. To address this concern, we revised our discussion to address a possible role for secreted factors in regulating sensory organ morphogenesis. Future studies will be aimed at learning more about the origin and cellular targets of shed EVs, and the mechanisms that stimulate EV shedding to provide answers regarding the role of EVs in ciliated neuron-glia communication.

1) Quoting from Jensen et al., 2016, 'sensory pore structure and function appears grossly normal in the rab-28 deletion mutant, which is likely a null allele'. The gk1040 allele was studied in this past publication. Yet, in the current manuscript, the same gk1040 allele and the additional allele tm2636 (which removes a smaller portion of Rab-28 than gk1040) both exhibit pronounced defects at the level of the sheath cells and the amphid pore. Could the authors comment on this discrepancy in the Discussion section?

There appears to be some confusion. Our original submission did not report a disrupted amphid sensory organ phenotype in *rab-28(tm2636)* worms. What we reported was an enlarged cephalic pore lumen and sheath cell defect in these worms. Perhaps the source of the confusion was a less-than-optimal presentation of our data. For this reason, we have made major changes in how the revised manuscript is organized, taking great care to ensure full clarity and distinction of cephalic and amphid organ phenotypes. The revised manuscript also now reports on *tm2636* amphid sensory organ ultrastructure. Consistent with what we reported in 2016 for *rab-28(gk1040)*, the *tm2636* lumen size appears grossly normal when compared to the control (see Figure 6 and Table 1). Thus, for amphid compartment size, we have no evidence of any difference or discrepancy between the *gk1040* and *tm2636* alleles. It is notable, however, that some abnormal matrix filled vesicles (MFV) occur in the amphid sheath of *tm2636* worms (see new data in Figure 6—figure supplement 1), which is something we did not see in the *gk1040* allele. There are two possible explanations for this distinction: (1) our analyses of *tm2636* (current study) and *gk1040* (2016 study) was done in males and hermaphrodites, respectively, and thus we cannot exclude sex distinctions in the matrix filled vesicle phenotype, and (2) unlike *gk1040*, the *tm2636* alelle is likely not a null, based on RTPCR analysis of *tm2636* transcript (see new data in Figure 4—figure supplement 1). This allele distinction could explain, at least in part, the difference in the sheath cell phenotype. All of the above is now clearly outlined in the revised manuscript. For clarity, we also now include Table 1 that summarizes the TEM and non-TEM phenotypes in all relevant genetic backgrounds.

2) The authors propose that Rab-28 functions downstream of the BBSome. Have the authors tested the localization of the BBSome in Rab-28 mutants? This experiment is particularly relevant in light of the sequence similarity between Rab-28 and RabL4/IFT27 and prior findings in mammalian cells that the BBSome accumulates in cilia when ift27 is deleted. The model that Rab-28 functions downstream of the BBSome in the regulation of EV shedding is based on the observations that Rab-28 localization to cilia and the PCMC is reduced in BBSome mutants (current manuscript and Jensen et al., 2016) while localization of the BBSome remains normal in the rab-28(gk1040) mutant (Jensen et al., 2016). In light of the newly found phenotypes of Rab-28 mutants in the current paper, it would be important to re-examine the localization and dynamics of the BBSome in Rab-28 mutants and in strains expressing Rab-28 variants.

We thank the reviewers for raising this important point. As suggested, we re-examined BBS-8 localization in non-EVN (amphid and phasmid) and EVN neurons of both *rab-28* mutant alleles. In amphid and phasmid neurons, we found that BBS-8::GFP ciliary localisation is unaffected in the *tm2636* and *gk1040* alleles, the latter validating previous findings from Jensen et al., 2016. In EVNs, there is some evidence of a modest reduction in ciliary BBS-8::GFP levels in *tm2636* but not in *gk1040*, possibly due to the allele differences described above. Nonetheless given technical challenges in measuring ciliary BBS-8::GFP levels in EVN cilia, and the fact that any defect is likely to be relatively small and restricted only to one allele, we do not have sufficient grounds to imply any role for RAB-28 in regulating BBSome localisation in a subset of cilia (i.e. those of EVNs). Thus, at this point, our conclusion that the BBSome regulates RAB-28 localization in non-EVNs, but not vice versa, stands.

**Author response image 1. respfig1:** BBS-8::GFP localization in *rab-28(tm2636)* Confocal Z-projections of BBS-8:: GFP in *rab-28(tm2636)* in amphid cilia (**A**), phasmid cilia (**B**), and the EV releasing ray neuron cilia (**C**). BBS-8:: GFP levels in *rab-28(tm2636)* are slightly reduced in ray neurons but, remain unaltered in amphid and phasmid cilia. Scale bar is 1μm for amphid and phasmid panels and 5μm for ray neuron cilia.

3) The authors propose that the EVs that fill the amphid pore lumen in bbs and rab-28 mutants originate from budding events at the periciliary membrane (microvesicle shedding) rather than from secretion of multivesicular bodies (exosome release). Have the authors looked at the distribution of CD63-GFP, a marker of ILVs and exosomes, in the rab-28 and bbs mutants? This is important. Is the release of EVs found in the amphid pore dependent upon ESCRT function?

Our genetic and electron microscopy data are consistent with these small EVs being ectosomes/microvesicles. We observed normal EV biogenesis, shedding and release phenotypes in ESCRT pathway mutants (Wang et al., 2014). With confocal microscopy and electron microscopy and tomography, we observe EVs in the lumen, suggesting that EVs are shed at the ciliary base into the lumen. Using EM, we do not see multivesicular bodies in the region of the ciliary base. Hence, these small ciliary EVs (average diameter 104.7 ± 46.7 nm) are not likely exosomes. Instead, we observe omega-shaped vesicle budding from the ciliary base suggesting that these *C. elegans* ciliary EVs are microvesicles (Wang et al., 2014; Silva et al., 2017). We now clearly state this in the revised Discussion section. In the final conclusion paragraph, we also note however, that we cannot exclude that some of the observed EVs could be of exosomal identity.

In the non-EVN amphid lumen, the ectopic (pathological) EVs observed in the bbs-8 mutants are poorly characterized in terms of content, regulation, and function, and a determination of the mode of EV biogenesis in the amphid sensory organ cannot be made until we learn of the identity of the cargoes within. For these reasons we have not made any conclusions regarding the origin of the ectopic EVs in the amphid sensory compartment.

Finally, in relation to the CD63 query, there is indeed an orthologue in *C. elegans (tsp-7)* as well as 21 other tetraspanin encoding gene paralogs. While CD63 is a common exosomal marker, a major challenge in the EV biology field is a dearth of markers that are specific for exosomes and for microvesicles/ectosomes. Most EV studies have used cultured cells or biofluids, and much effort has been invested in improving methodology for isolating and characterizing different EV subtypes using protein and lipid markers. Exosomes (30-150 nm in diameter) are generated through fusion of multivesicular bodies. Microvesicles or ectosomes (100-1000 nm) bud from the plasma membrane. EVs are heterogeneous and typically isolated based on size and density and are of unknown biogenic origin: available markers are not strictly specific for exosomes or microvesicles/ectosomes. Therefore, the terms small EVs (<200 nm) and large EV (>200 nm) are used (Maas, Breakefield and Weaver, (2017); Meldolesi, (2018). Hence the term “small EV” encompasses both exosomes and small ectosomes/microvesicles (Thery et al., 2018). For all of these reasons, we are conservative in our diagnosis of exosome versus ectosome, and use the term “EVs.” We have added excerpts of this paragraph to the Introduction.